# *Candida albicans* exhibits heterogeneous and adaptive cytoprotective responses to antifungal compounds

Vanessa Dumeaux[1], Samira Massahi[2], Van Bettauer[3], Austin Mottola[4], Anna Dukovny[4], Sanny Singh Khurdia[2], Anna Carolina Borges Pereira Costa[2], Raha Parvizi Omran[2], Shawn Simpson[3], Jinglin Lucy Xie[5], Malcolm Whiteway[2], Judith Berman[4], Michael T Hallett[6]*

[1]Department of Anatomy and Cell Biology, Western University, London, Canada; [2]Department of Biology, Concordia University, Montreal, Canada; [3]Department of Computer Science and Software Engineering, Concordia University, Montreal, Canada; [4]Shmunis School of Biomedical and Cancer Research, The George S. Wise Faculty of Life Sciences, Tel Aviv University, Tel Aviv-Yafo, Israel; [5]Department of Chemical and Systems Biology, Stanford University, Stanford, United States; [6]Department of Biochemistry, Western University, London, Canada

*For correspondence:
michael.hallett@uwo.ca

Competing interest: The authors declare that no competing interests exist.

**Abstract** *Candida albicans*, an opportunistic human pathogen, poses a significant threat to human health and is associated with significant socio-economic burden. Current antifungal treatments fail, at least in part, because *C. albicans* can initiate a strong drug tolerance response that allows some cells to grow at drug concentrations above their minimal inhibitory concentration. To better characterize this cytoprotective tolerance program at the molecular single-cell level, we used a nanoliter droplet-based transcriptomics platform to profile thousands of individual fungal cells and establish their subpopulation characteristics in the absence and presence of antifungal drugs. Profiles of untreated cells exhibit heterogeneous expression that correlates with cell cycle stage with distinct metabolic and stress responses. At 2 days post-fluconazole exposure (a time when tolerance is measurable), surviving cells bifurcate into two major subpopulations: one characterized by the upregulation of genes encoding ribosomal proteins, rRNA processing machinery, and mitochondrial cellular respiration capacity, termed the Ribo-dominant (*Rd*) state; and the other enriched for genes encoding stress responses and related processes, termed the Stress-dominant (*Sd*) state. This bifurcation persists at 3 and 6 days post-treatment. We provide evidence that the ribosome assembly stress response (RASTR) is activated in these subpopulations and may facilitate cell survival.

## Editor's evaluation

The valuable study by Dumeaux et al. examines the transcriptional response to antifungal treatment in the major opportunistic human fungal pathogen *Candida albicans*. Using solid methodology, including a novel droplet-based single cell transcriptomics platform, the authors report that fungal cells exhibit heterogeneity in their transcriptional response to antifungal drug treatment. The ability to study the trajectories of individual cells in a high-throughput manner provides a novel perspective on studying the emergence of drug tolerance and resistance in fungal pathogens.

**eLife digest** Many drugs currently used to treat fungal diseases are becoming less effective. This is partly due to the rise of antifungal resistance, where certain fungal cells acquire mutations that enable them to thrive and proliferate despite the medication. Antifungal tolerance also contributes to this problem, wherein certain cells can continue to grow and multiply, while other – genetically identical ones – cannot. This variability is partly due to differences in gene expression within the cells. The specific nature of these differences has remained elusive, mainly because their study requires the use of expensive and challenging single-cell technologies.

To address this challenge, Dumeaux et al. adapted an existing technique to perform single-cell transcriptomics in the pathogenic yeast *Candida albicans*. Their approach was cost effective and made it possible to examine the gene expression in thousands of individual cells within a population that had either been treated with antifungal drugs or were left untreated.

After two to three days following exposure to the antifungal treatment, *C. albicans* cells commonly exhibited one of two states: one subgroup, the 'Ribo-dominant' cells, predominantly expressed genes for ribosomal proteins, while the other group, the 'Stress-dominant' cells, upregulated their expression of stress-response genes. This suggests that drug tolerance may be related to different gene expression patterns in growing cell subpopulations compared with non-growing subpopulations.

The findings also indicate that the so-called 'ribosome assembly stress response' known to help baker's yeast cells to survive, might also aid *C. albicans* in surviving exposure to antifungal treatments.

The innovative use of single-cell transcriptomics in this study could be applied to other species of fungi to study differences in cell communication under diverse growth conditions. Moreover, the unique gene expression patterns in *C. albicans* identified by Dumeaux et al. may help to design new antifungal treatments that target pathways linked to drug resistance.

## Introduction

*Candida albicans* is one of the most prevalent human fungal pathogens (*Benedict et al., 2019*; *Kullberg and Arendrup, 2015*; *Pappas et al., 2018*). Systemic *C. albicans* infections are the second most common cause of mortality from infectious diseases in extremely premature infants (15–20% mortality), and the fourth most common cause of nosocomial bloodstream infections (30–50% mortality) (*Benjamin et al., 2010*; *Brown and Netea, 2012*; *Pfaller et al., 2019*). The frequency of drug resistance is far lower than the frequency of treatment failures, such that most infections that fail to respond to antifungal drug treatments are susceptible to the drug. This clinical persistence can result from heterogeneity in antifungal drug responses, variations in host immune status, and the inability of drugs to reach their fungal target sites (*Delarze and Sanglard, 2015*; *Wuyts et al., 2018*).

A *resistant* fungal isolate can thrive in the presence of an antifungal drug at concentrations exceeding the minimum inhibitory concentration (MIC) of that species, with detectable growth within 24 hr of drug exposure. For a *tolerant* fungal isolate, a proportion of cells (typically 5–95% of the population) can grow, albeit slowly, at concentrations surpassing the population average MIC (*Berman and Krysan, 2020*); tolerance is detectable only 48 hr after drug exposure, which is longer than most clinical assays are performed (*Espinel-Ingroff et al., 2017*). Approaches for predicting patient response to an infection, based solely on the average response of cells to a drug challenge across a population after 24 hr, can overlook such subpopulations with behaviors that have the potential to significantly impact the evolution of drug resistance (*Altschuler and Wu, 2010*).

Resistance usually results from mutations that directly influence interaction of the drug with its target (*Berman and Krysan, 2020*), while tolerance is thought to stem from phenotypic heterogeneity or cell-to-cell variations in phenotypic responses within an isogenic cell population. Microbial phenotypic heterogeneity can arise from mechanisms including stochastic or periodic gene expression, protein stability, cell age, cell-cell interactions, chromatin modifications, and genomic neoplasticity (*Ackermann, 2015*). Heterogeneity in microbial populations can confer benefits like bet-hedging, rapid metabolic shifts, division of labor, and resource sharing (*Ackermann, 2015*). It is important to note the distinction between antibacterial and antifungal drug tolerance. For bacteria, tolerance is defined by the duration of cell survival following periodic exposure to a bactericidal drug (*Brauner et al., 2016*). By contrast, for fungi, tolerance is characterized by the proportion of growth

in supra-MIC drug concentrations relative to growth without the drug and is best characterized in *C. albicans* treated with fluconazole (FCZ). The underlying mechanisms and implications of antifungal drug tolerance differ from those of bacterial heteroresistance, in which a small subpopulation exhibits growth in bacteriocidal drugs (*Andersson et al., 2019*). In this study, the focus is on antifungal tolerance to FCZ, a well-known fungistatic drug.

Previous studies have characterized key aspects of drug tolerance in *C. albicans*. Tolerance is largely drug concentration-independent and high tolerance correlates with poor clinical outcomes (*Astvad et al., 2018*; *Levinson et al., 2021*; *Rosenberg et al., 2018*). Tolerance increases the effective population size (*Cowen and Lindquist, 2005*; *Vincent et al., 2016*). The emergence of higher tolerance cannot be attributed solely to the accumulation of adaptive point mutations (*Wertheimer et al., 2016*), as selected tolerant cells and their original parent give rise to mixed populations of tolerant and non-tolerant cells in similar proportions (*Rosenberg et al., 2018*). Adjuvant drugs used alongside the common fungistatic FCZ not only inhibit tolerance but prevent the evolution of resistance because these drug combinations are fungicidal. Such adjuvant drugs can modulate the tolerance response by targeting Hsp90, calcineurin, TOR, PKC, and sphingolipid biosynthesis (*Cowen and Lindquist, 2005*; *Karababa et al., 2006*; *Rosenberg et al., 2018*; *Sanglard et al., 2003*; *Vincent et al., 2016*). The molecular mechanisms coordinating tolerance regulation across cellular pathways are not yet fully understood, although stress responses appear to be central (*Cowen and Lindquist, 2005*; *Rosenberg et al., 2018*). Several cellular processes are essential for tolerance, including ergosterol biosynthesis and proteasome function (reviewed in *Berman and Krysan, 2020*).

Most previous studies of drug tolerance, ranging from classic growth assays to omics profiling, have treated surviving fungal subpopulations as a homogenous whole, mainly due to a lack of technology to efficiently study large numbers of individual fungal cells. While single-cell (sc) transcriptomic assays have been applied extensively to mammalian systems, their use in fungal contexts remains limited due to the challenges of disrupting the rigid cell wall, lysing the membrane, and the small overall amount of RNA per cell. In the model yeast *Saccharomyces cerevisiae*, several limited microfluidic or barcoding sc studies examined approximately a hundred cells (*Gasch et al., 2017*; *Nadal-Ribelles et al., 2019b*; *Nadal-Ribelles et al., 2019a*; *Urbonaite et al., 2021*), and two high-throughput fungal transcriptome studies profiled ~40K cells (*Jackson et al., 2020*; *Jariani et al., 2020*). In our preliminary pre-printed study (*Bettauer et al., 2020*), we profiled *C. albicans* using a fungal nanoliter droplet-based assay (DROP-seq), modified from the original system presented by *Macosko et al., 2015*. This approach overcomes technical challenges that arise in fungal settings, providing a flexible, cost-effective solution. Here, we extend the data and analysis from Bettauer et al. to better understand subpopulation-specific responses to drug stress. Since the original sc study (*Bettauer et al., 2020*), a second *C. albicans* sc study was performed to investigate the initial response (within 3 hr) to high FCZ concentrations (*Dohn et al., 2022*). Dohn and colleagues observed increased expression of ergosterol genes, an increase in cell cycle arrest genes, and a transient increase in stress response gene expression at 1.5 hr. The work also identifies interesting variability in the temporal response to acute drug exposure but does not address molecular heterogeneity in the subsequent emergence from cell cycle arrest that can only be observed as tolerance after longer times (days 2–3), the focus of this study.

Here, we use the fungal DROP-seq system (*Bettauer et al., 2020*) to explore phenotypic heterogeneity in the *C. albicans* response to antifungal drugs. We profiled the transcriptomes of thousands of individual cells from *C. albicans* populations that were either untreated (UT) or exposed to antifungal compounds with a focus on FCZ across several days. This data is integrated with bulk DNA sequencing and fluorescence microscopy to provide in-depth analyses that focus on subpopulation composition and phenotypic heterogeneity across isogenic cell populations. We detect a heterogeneous response to FCZ within isogenic populations, supporting the idea that different cells exhibit distinct survival strategies, including some with increased expression of genes related to drug tolerance. This study underscores molecular events that may lead to drug tolerance and that hold potential for future therapeutic targeting.

## Results

### *C. albicans* exhibits drug tolerance 2 days after exposure to FCZ

To uncover molecular events associated with the emergence of drug tolerance in *C. albicans*, we performed a series of disk diffusion assays, focusing on a 6-day time series with FCZ, detailed in Appendix 1. When assayed at 48 hr, both disk assays and broth microdilution assays report on tolerance (*Rosenberg et al., 2018*). Drug tolerance increased significantly between days 1 and 2 (p<0.001, Kruskal-Wallis $\chi^2$ test) and increased slightly at day 3, after which the tolerance level did not change. Thus, tolerant cells are present in all cultures starting at day 2. Since these populations originate from a single isogenic colony, we assume that tolerant and non-tolerant cells in the same culture differentially express pathways and processes relevant to their ability to grow (or not grow) in the presence of the drug. We do not observe widespread drug resistance at any time point.

### An optimized sc profiling assay to explore drug tolerance in *C. albicans*

Although sc profiling with a commercial system is feasible in *S. cerevisiae* (*Jackson et al., 2020*), specific aspects of fungal biology motivated us to develop a low-cost alternative tailored for fungi, specifically *C. albicans*. We first optimized the removal of the cell wall, as well as the time and concentration parameters to fix the transcriptome (Materials and methods, 'Strains, media, and drug

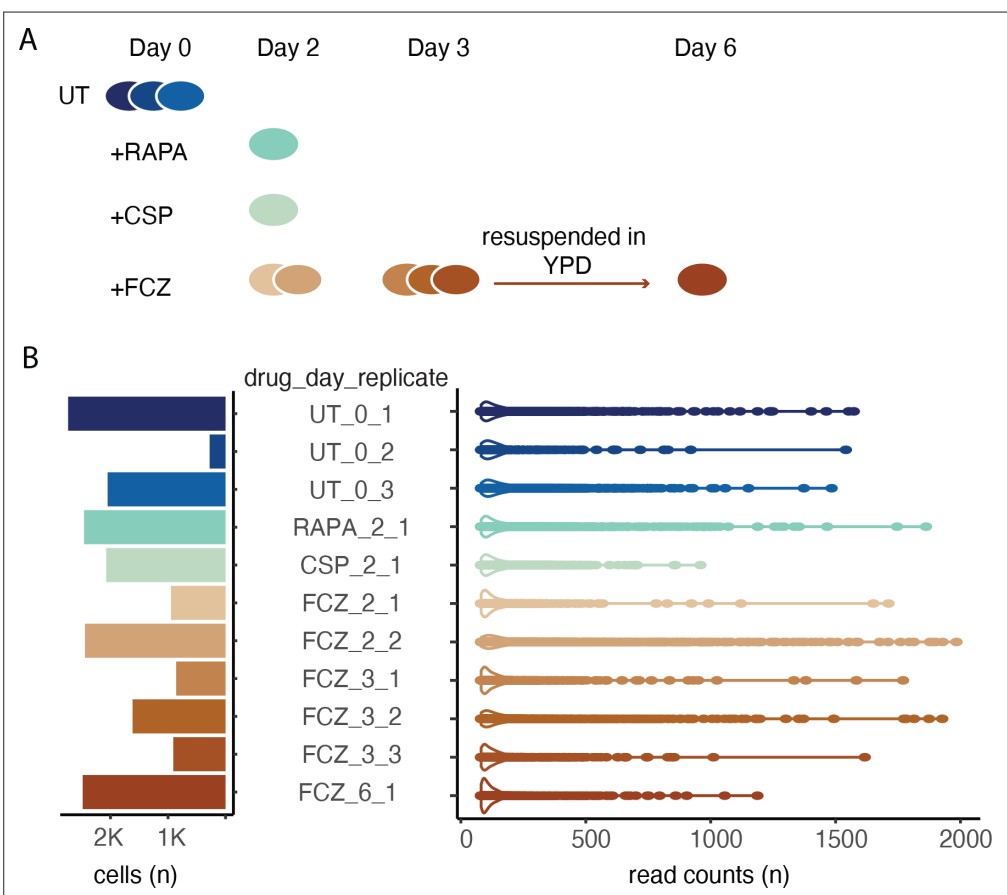

**Figure 1.** Experimental design and initial single-cell profiling. (**A**) The time series experiment begins with three replicates of untreated (UT) cells followed by profiling of rapamycin (RAPA), caspofungin (CSP), and fluconazole (FCZ, 2 replicates at day 2, and 3 replicates at day 3). After 3 days in FCZ, cells were transferred to YPD; recovered cells were profiled at day 6 (i.e., 3 days after resuspension). (**B**) Bar plot (left) depicts the number of high-quality cells per sample. Violin plots (right) the distribution in the number of reads assigned to each cell.

The online version of this article includes the following figure supplement(s) for figure 1:

**Figure supplement 1.** Schema for analysis stragegy and measures of gene experssion.

**Figure supplement 2.** Comparison of single cell and bulk expression profiles.

treatment', 'Spheroplasts'). We then built a nanoliter droplet-based system modified from *Macosko et al., 2015*, using in-house components as described previously (*Booeshaghi et al., 2019*), to reduce the device and assay costs. The profiles reported here combine our preliminary effort (*Bettauer et al., 2020*) with additional data and analyses to provide increased power to examine the technical and biological efficacy of our system.

*C. albicans* populations were grown in rich media alone (UT) or with an antifungal compound: FCZ (1 µg/ml), caspofungin (CSP; 1 ng/ml), or rapamycin (RAPA; 0.5 ng/ml) (Materials and methods, 'Strains, media, and drug treatment'). UT samples were collected during the logarithmic growth phase and treated samples were collected at days 2 and 3 post-drug exposure. This is the period when FCZ tolerance becomes evident. To explore whether subpopulations observed at days 2 and 3 persisted at later time points, we resuspended the FCZ day 3 population in fresh medium and profiled the samples on day 6 after 3 days of growth without drug (*Figure 1A*). The vast majority of cells were in the yeast white morphology with less than 0.2% of cells exhibiting a filamentous morphology (hyphae or pseudohyphae).

After processing with the DROP-seq device, samples were sequenced following the original protocol (*Macosko et al., 2015*) but with cell concentration and PCR cycle numbers optimized for *C. albicans* (Materials and methods, 'Cell preparation for sc profiling'). Sc sequence data was processed using a reference index that covers the spliced transcriptome and the Alevin-Fry package (*He et al., 2021*) (Materials and methods, 'Quality control, basic processing, and normalization of the sc profiles'). Gene and cell quality control are challenging exercises in all sc profiling efforts (*Svensson, 2019*), and especially in fungi because of the small amount of RNA per cell, especially under the stress of growth in antifungal drugs (*Jariani et al., 2020*). A series of quality control procedures were used to estimate gene/cell expression levels (*Figure 1—figure supplement 1A*, Materials and methods, 'Quality control, basic processing, and normalization of the sc profiles').

The pipeline identified 18,854 high-quality cells across the 11 drug/time point conditions with an average of 1714 cells per sample. On average, 184 transcripts were identified in each cell, however there is large dispersion in the right tail representing many cells with significantly more transcripts (max. 1984) (*Figure 1B*). On average, these transcripts arose from 94 unique genes per cell, again with large right tail dispersion (max. 825) (*Figure 1—figure supplement 1B and C*). Since theoretical results highlight the importance of many cells over the number of identified genes per cell (*Zhang et al., 2020*), we reasoned that inclusion of the sparse cells (left tail) would strengthen the analyses and help identify large subpopulations with strong differentially expressed transcriptional programs across the different treatments. Moreover, although the gene by cell count matrix was sparse, there was very high concordance between FCZ pseudo-bulk profiles (i.e., aggregated sc counts) at day 2 and day 3 (R=0.82; *Figure 1—figure supplement 2A*), indicating that the assay robustly quantifies the expression of genes across different batches.

To further investigate the robustness of the assay, we performed bulk RNA-sequencing of FCZ-treated cells at day 2 post-treatment (Materials and methods, 'Bulk transcriptomics') and compared this bulk profile with the pseudo-bulk derived from mapping sc reads to the reference genome but ignoring barcodes (unfiltered 'pseudo-bulk' profiles; Materials and methods, 'Construction of pseudo-bulk profiles'). The comparison identified 6071 genes with only 172 genes not detected in one or more of the pseudo-bulk profiles. We note that missing genes were mostly lowly expressed genes in the bulk profile (*Figure 1—figure supplement 2B*). Moreover, day 2 and day 3 pseudo-bulk replicates were significantly correlated with the bulk RNA-sequencing (*Figure 1—figure supplement 2C*). This strongly suggests that the DROP-seq-derived profiles sample the *C. albicans* transcriptome, capture true biological signals, and primarily miss transcripts expressed at lower levels.

## In isogenic UT cells, differences in cell transcriptional profiles highlight metabolic and stress responses coupled with cell cycle checkpoints

As described above, mid-log phase cells grown under standard conditions were collected for sc profiling. In addition to sc transcriptomes, 'bulk' DNA-sequencing profiles were generated to verify strain isogenicity (Materials and methods, 'Whole-genome DNA-sequencing').

To identify the main sources of cell-to-cell variability in UT cells (N=5062 cells), we compiled gene signatures for biological processes and responses likely to play a role in microbial phenotypic

heterogeneity based on previous transcriptional profiling studies. Gene signatures often consist of genes that are differentially expressed when a specific process is activated (or repressed) compared to wildtype cells. Here, the signatures included processes such as cell cycle, stress responses (both specific and general), the TCA cycle, and metabolic pathways (such as glycolysis) (*Figure 2—source data 1*, Materials and methods, 'Cell clustering, trajectory, and signature analyses'). Most of these expression signatures were derived in the context of bulk transcriptional studies either directly from *C. albicans* or from other fungi. Where necessary, we identified orthologs of relevant genes in *C. albicans* (*Balakrishnan et al., 2012*), while considering regulatory differences between species (*Johnson, 2017*; *Lavoie et al., 2010*).

The score of each signature was measured in each UT cell profile (Materials and methods, 'Cell clustering, trajectory, and signature analyses'); the most variable signatures are displayed in *Figure 2A and B*. One large group of cells exhibited elevated expression of genes involved in both M phase and the heat-shock response (cell indices >3000), consistent with a study linking these two processes (*Senn et al., 2012*). We also found that a small group of cells with very high expression of heat-shock response exhibited the lowest expression of M and S phase, glycolysis, and RP-coding-related genes (cell indices ~2700–3000). These few cells likely experienced cell cycle arrest due to high levels of cellular stress. Most other M phase cells with relatively high expression of heat-shock protein-coding genes (indices 3000–5000) also exhibited high expression of glycolysis-related genes. In contrast, cells with the least evidence of M phase expression, sometimes exhibited high S phase gene expression, low expression of the heat-shock signature, and relatively high expression of the oxidative stress signature (cell indices 0–2700, *Figure 2A and B*). Overall, these results indicate that expression heterogeneity in UT cells is primarily related to different cell cycle phases, distinct stress responses, and metabolic states. Of note, the UT cells were grown in rich media without exposure to any known stressors.

To further investigate the hypothesis that UT cells differentially express genes involved in specific stress responses, we chose gene pairs predicted to have mutually exclusive expression in any given cell based on the sc transcriptomics profiles; for example, heat-shock protein 70 (HSP70) and dithiol glutaredoxin (TTR1), which is involved in the oxidative stress response (*Figure 2C*; McNemar test, p-value <0.001). We constructed a dual fluorescent reporter strain expressing GFP-labeled Hsp70 and RFP-labeled Ttr1 (Materials and methods, 'Cell imaging'). During growth in rich medium, fluorescence microscopy revealed a notable level of expression with only one of the two markers detectable in a given cell as predicted from the sc profiles (*Figure 2D, E*, *Figure 2—figure supplement 1*, McNemar test, p-value <0.001). Thus, the distinct transcriptional stress responses observed in the UT population data are not due to stress responses during sc profiling. Instead, they most likely represent the cell cycle stage and/or the metabolic condition of the cell, emphasizing the significance of metabolic- and stress-sensitive phases in the progression of the cell cycle. These results align with several previous reports that associate cell cycle phase with the expression of stress response and metabolism-related genes (*Brauer et al., 2008*; *Chiu et al., 2011*; *Hossain et al., 2021*; *Senn et al., 2012*).

## Cells display distinct survival responses to FCZ

We next investigated the *C. albicans* response to antifungal compounds using fungistatic FCZ (1 µg/ml, 1–2 × MIC$_{50}$, *Figure 3—figure supplement 1A*), and fungicidal CSP (1 ng/ml, <0.03 × MIC). These concentrations were chosen to ensure a sufficient number of survivors. FCZ targets Erg11p, which encodes an enzyme central to ergosterol biosynthesis and membrane integrity (*Odds et al., 2003*; *Thamban Chandrika et al., 2018*; *Wertheimer et al., 2016*). At these concentrations, FCZ treatment slows growth relative to UT controls in the first days after exposure (*Figure 3—figure supplement 1B*). Since tolerance to FCZ should be evident after 2 days of incubation in the presence of the drug (*Gerstein and Berman, 2020*), we profiled our *C. albicans* cell subpopulations at 2 and 3 days post-exposure using drug disk assays (Appendix 1). Finally, we also investigated the *C. albicans* response to low doses of RAPA (MIC$_{80}$ <1 µg/ml, 0.5 ng/ml; 0.0005× MIC) (*Cruz et al., 2001*). RAPA is known to inhibit tolerance to FCZ (*Rosenberg et al., 2018*).

We focused on 11,309 good-quality cells captured at 2 or 3 days post-exposure to FCZ, CSP, and RAPA, in addition to the 5062 UT cells. Unsupervised analysis was used to identify similar clusters of cells. The analysis first uses a deep generative neural network (scVI) to transform the high-dimensional expression profiles down to lower dimensions, followed by Leiden clustering to identify groups of cells that have similar expression profiles (Materials and methods, 'Quality control, basic processing, and

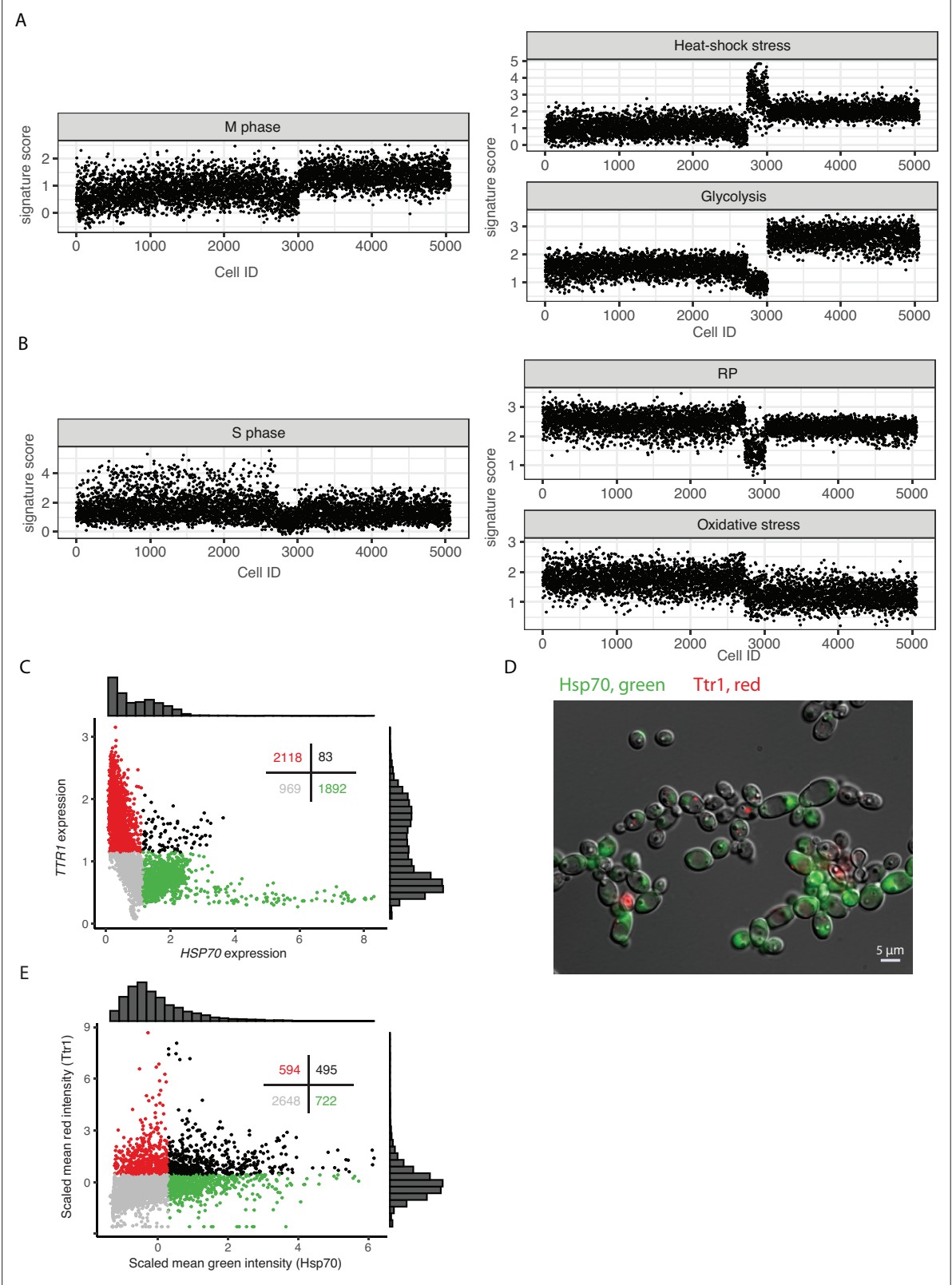

**Figure 2.** Cell-to-cell heterogeneity in untreated (UT) cell populations. (**A**, **B**) Expression levels (VISION scores) of curated signatures for individual UT cells. Cell order is the same in all graphs. (**C**) Scatterplot of cells (dots) based on expression level after imputation. Colors indicate: red, *TTR1* expression >1.2 and *HSP70* expression <1.2; green, *TTR1* expression <1.2 and *HSP70* expression >1.2; black, both *TTR1* and *HSP70* expression >1.2; and gray, expression of both genes was <1.2. Distributions of expression are illustrated in histograms above and to the right and the number of cells

*Figure 2 continued on next page*

*Figure 2 continued*

in each group is provided in the top right of the figure. (**D**) Representative fluorescence micrographic image of RFP-tagged TTR1(red) and GFP-tagged HSP70 (green) in a population of isogenic cells showing the mutually exclusive nature of their expression in individual cells. (**E**) Plot of the mean intensities captured in microscopy images of RFP-tagged TTR1 (red) and GFP-tagged HSP70 (green).

The online version of this article includes the following source data and figure supplement(s) for figure 2:

**Source data 1.** Gene signatures related to microbial phenotypic diversity and drug tolerance curated from the literature.

**Source data 2.** GFP and RFP oligo primers.

**Figure supplement 1.** Examples of microscopy images of *C. albicans* cells harboring both a GFP-tagged heat-shock protein 70 (HSP70) and an RFP-tagged TTR1.

normalization of the sc profiles', 'Cell clustering, trajectory, and signature analyses'). This revealed five main clusters containing 92% of all N=16,371 cells, in addition to ~20 small clusters. We used Uniform Manifold Approximation and Projection (UMAP) to visualize these patterns in 2D; cells are colored by their cluster assignment (***Figure 3A***). To investigate the resilience of the cell clusters, we randomly repeated the clustering process 100 times, each time using a random subset consisting of 95% of the cells. Cells originally assigned to four of the main clusters almost always remained in that cluster (***Figure 3—figure supplement 1C***). A significant portion of cells allocated to the (smaller) cluster 5 (5-purple, N=501 cells, 3%) were categorized together with cluster 3 cells, which is indicative of only subtle gene expression differences and suggests the two clusters are collapsed (***Figure 3—figure supplement 1C***). The most variable signatures across clusters are displayed in ***Figure 3C***, where color corresponds to the average score (z-score, color bar) across all cells within each cluster. UT cells are primarily found in cluster 3-green and to a lesser extent in cluster 2-pink (***Figure 3B***). CSP cells were mostly found in cluster 1-darkpink while RAPA cells were mostly found in cluster 2-lightpink (***Figure 3B***). Biological inferences associated with each of these clusters are further described in Appendix 2. The remaining 3% of cells (N=573) were outliers scattered across ~20 distinct clusters ('comet'-like cluster-darkblue; ***Figure 3A***). The 'comet-like' clusters appear in random directions from the five main clusters. This pattern suggests that the small set of cells in each comet have strong transcriptional similarity, but each such comet is transcriptionally distinct from other comets. The comets were primarily observed in FCZ survivors at 2 days (***Figure 3B***); an investigation of the relevant biology underlying comets is discussed in Appendix 2.

Interestingly, a clear bifurcation in the FCZ survivor cell population was evident in the sc transcriptional profile by day 2 (***Figure 3D***), corresponding to clusters 1-darkpink and 4-turquoise of ***Figure 3A***. We term these two distinct states the Ribo-dominant (*Rd*) state and the Stress-dominant (*Sd*) state, respectively. The bifurcation between the *Rd* and *Sd* states becomes more pronounced by day 3 (***Figure 3E***), with nearly every surviving FCZ-treated cell appearing in either the *Rd* or *Sd* state.

The *Rd* state is characterized by high RP gene expression, moderate to high expression of GCN4-mediated response genes that activate amino acid biosynthesis, along with low expression of glycolytic and carbohydrate reserve metabolic pathway genes, and an absence of heat-shock or hyperosmotic stress response genes (***Figure 3C-G***; ***Figure 3—figure supplement 2A***). By contrast, the *Sd* state is characterized by high expression of heat-shock stress response genes and low expression of RP genes (***Figure 3C-G***; ***Figure 3—figure supplement 2A***).

In total, 797 genes are differentially expressed between the *Rd* and the *Sd* state (pseudo-bulk from cluster FCZ 1-dark pink versus cluster 4-turquoise; DESeq2, FDR <0.1; Materials and methods, 'DGE analysis'; ***Figure 3—source data 1*** ). Among these genes, 230 are overexpressed in *Rd* cells, with half of them involved in protein translation (***Figure 3—source data 1 and 2***; ***Figure 3—figure supplement 2B***). Many highly expressed genes within the *Rd* response were also involved in rRNA processing and mitochondrial cellular respiration.

By contrast, 567 genes are more highly expressed in the *Sd* state; this set of genes is enriched for involvement in cell wall organization, cell adhesion, morphology, virulence, filamentous growth, and biofilm formation (***Figure 3—source data 1 and 2***; ***Figure 3—figure supplement 2B***). Interestingly, *Sd* cells also highly express genes involved in the unfolded protein response (UPR), such as *HSP70* and *YHB1*, as well as genes that promote drug tolerance, such as *HSP90*, *GZF3*, *CCH1*, *HSP21*, *HSP70*, and *RIM101* (***Delarze et al., 2020***; ***Garnaud et al., 2018***; ***Liu et al., 2015***; ***Mayer et al., 2013***; ***Nagao et al., 2012***; ***Rosenberg et al., 2018***).

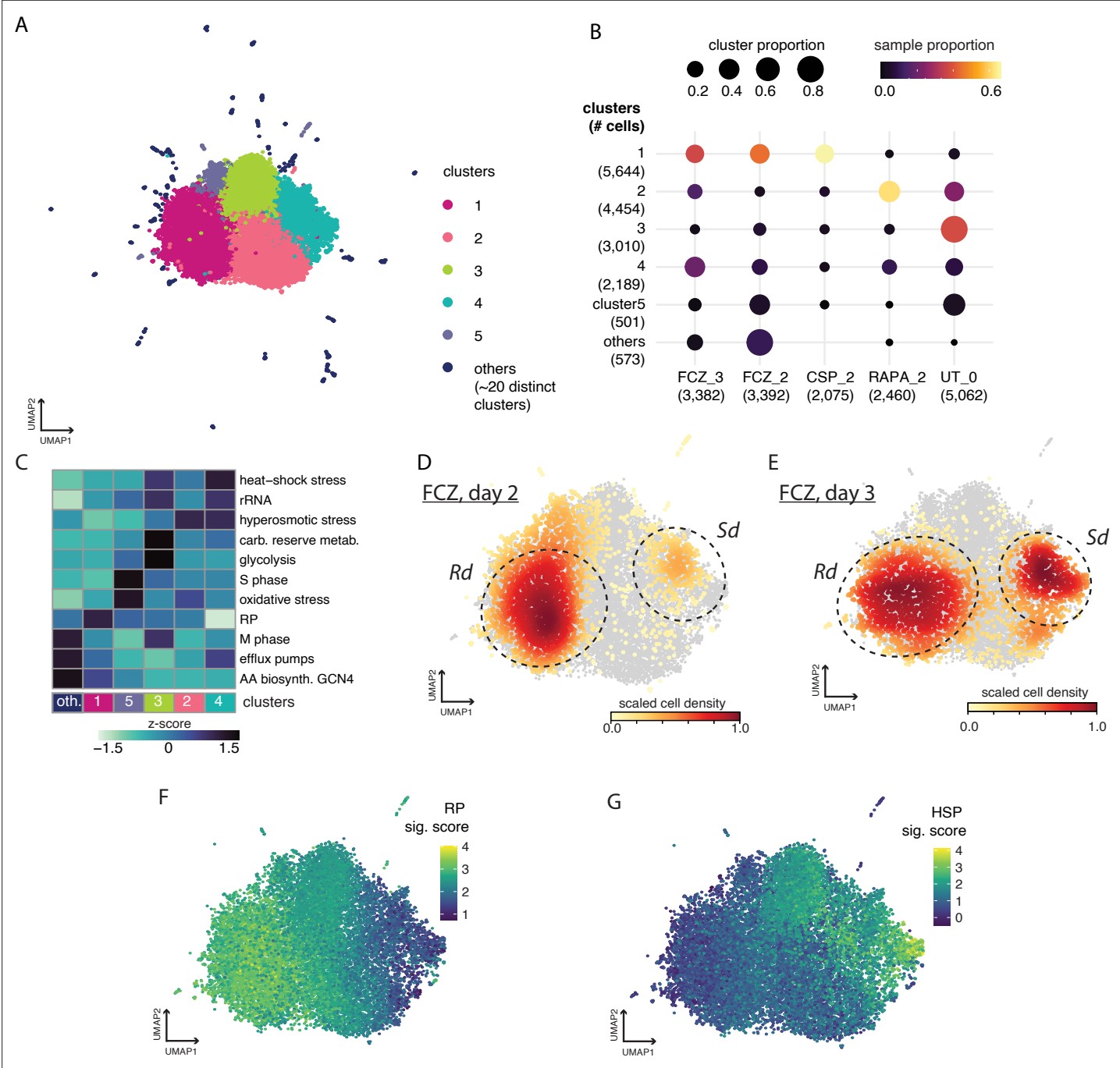

**Figure 3.** Cell profiles after challenges with different antifungal compounds. (**A**) Uniform Manifold Approximation and Projection (UMAP) embedding of all cells including untreated (UT) cells and cells treated with fluconazole (FCZ) (days 2 and 3), rapamycin (RAPA) (day 2), and caspofungin (CSP) (day 2). Leiden clustering identified five main clusters and ~20 sparsely populated 'comets'. (**B**) Dotplot describing relative size of cluster populations. Rows correspond to clusters and columns correspond to drug_day conditions. Dot diameter is proportional to the fraction of cells from each condition for a given cluster. The dot color is proportional to the fraction of cells from each cluster for a given condition as shown in the color bar. Numbers in parentheses indicate the total cell count in clusters and drug_day conditions. (**C**) A heatmap depicting the level of activation (VISION z-scores) of different signatures. (**D–G**) The UMAP embedding from panel (**A**) but where color depicts: (**D**) density of FCZ at day 2 cells, (**E**) density of FCZ at day 3 cells. (**F,G**) Signature scores for expression of (**F**) ribosomal protein (RP), (**G**) and heat-shock stress (HSP) signatures.

The online version of this article includes the following source data and figure supplement(s) for figure 3:

**Source data 1.** Differentially expressed genes in fluconazole (FCZ)-treated cells classified in the Ribo-dominant (Rd) cluster 1 (pseudo-bulk samples) compared to FCZ-treated cells in the Stress-dominant (Sd) cluster 4 (pseudo-bulk samples).

*Figure 3 continued on next page*

Figure 3 continued

**Source data 2.** Gene ontology terms enriched in genes differentially expressed between fluconazole (FCZ)-treated cells classified in the Ribo-dominant (Rd) cluster 1 (pseudo-bulk samples) compared to FCZ-treated cells in the Stress-dominant (Sd) cluster 4 (pseudo-bulk samples) listed in *Figure 3—figure supplement 2B*.

**Figure supplement 1.** Differential growth characteristics of strain SC5314 in response to FCZ.

**Figure supplement 2.** Pathway analysis between the Rd and Sd cell clusters.

The sc profiles revealed that isogenic cells that survived FCZ treatment are found in one of two distinct states at days 2 and 3. This highlights the heterogeneity of cellular responses to FCZ treatment, which may be associated with drug tolerance and clarifies that one state (*Rd*) is enriched in ribosome and translation-related functions while the other state (*Sd*) is enriched in genes related to genes identified in the stress responses including the UPR and drug tolerance.

## Drug tolerance in *C. albicans* may involve the ribosome assembly stress response to facilitate a transition from the *Rd* to *Sd* state

The *Rd* state, evident on days 2 and 3, is marked by increased expression of RP and rRNA processing genes. Proper ribosomal biogenesis requires a balance between the synthesis of RP and rRNA. An imbalance between RP and rRNA processing can lead to proteotoxic stress and RP aggregation (*Tye et al., 2019*). This stress and aggregation activate the ribosome assembly stress response (RASTR) in yeast, which involves heat-shock transcription factor Hsf1 (*Albert et al., 2019*). Hsf1 subsequently upregulates HSP90 and other protein folding-related genes, including HSP70, which are upregulated in the *Sd* state cells. In this context, we propose that the drug tolerance response in *C. albicans* may involve RP aggregation (in *Rd* cells), which activates RASTR, and induces the expression of Hsf1 (in *Sd* cells). This process might allow cells to transition from the *Rd* to *Sd* state, where genes implicated in drug tolerance are expressed.

To test this hypothesis, we collected a list of *C. albicans* orthologs of *S. cerevisiae* RASTR genes (gene list supplied by B Albert, *Table 1A*) together with a list of *C. albicans* genes which are targets of Hsf1 (*Leach et al., 2016*; *Table 1B*). In *S. cerevisiae,* the characteristic RASTR gene expression profile includes decreased expression of several RP genes and increased expression of Hsf1-regulated genes involved in protein folding, proteolysis, and reaction to heat (*Albert et al., 2019*). In our sc profiles, *Sd* cells exhibit increased expression of RASTR upregulated genes and constitutive targets of Hsf1 (*Figure 4A and B*), and we have previously shown (*Figure 3E–F*) that *Sd* cells exhibit decreased expression of ribosome processing and ribosome protein-encoding genes. This suggests that both

**Table 1.** (A) Orthologous genes in *C. albicans* associated with ribosome assembly stress response (RASTR) in *S. cerevisiae*. (B) Constitutive Hsf1 target genes.

**RASTR signature (Albert et al., 2019)**

| | | |
|---|---|---|
| DOWN | Ribosome processing | ASC1, RPL10, RPL10A, RPL11, RPL12, RPL13, RPL14, RPL16A, RPL17B, RPL18, RPL19A, RPL2, RPL20B, RPL21A, RPL23A, RPL24A, RPL25, RPL27A, RPL28, RPL30, RPL32, RPL35, RPL37B, RPL38, RPL39, RPL42, RPL5, RPL6, RPL8B, RPL9B, RPP0, RPP1B, RPS1, RPS10, RPS12, RPS13, RPS14B, RPS15, RPS16A, RPS17B, RPS18, RPS19A, RPS20, RPS21, RPS21B, RPS22A, RPS23A, RPS25B, RPS26A, RPS27, RPS27A, RPS3, RPS5, RPS6A, RPS7A, RPS8A, RPS9B, UBI3, YST1 |
| UP | Protein folding, response to heat, proteolysis | UBI4, RPN4, PIN3, STF2, KAR2, MSI3, HSP60, HSP90, HSP70, SIS1, SSA2, HSP104, HSP78, STI1 |
| | Glucose and pyruvate metabolic process | CYP1, PMA1, GLK1, TDH3, CDC19, PGK1 |
| | Unknown function | MBF1, KRE30, YDJ1, YBN5, ACT1, UBC4 |

**Hsf1 constitutive targets (*Leach et al., 2016*)**

ACE2, ADAEC, AHA1, ALO1, ALS1, ALS3, ALS4, ASR1, BOI2, BUL1, CCP1, CDC37, CDC48, CPR6, CRD2, CTF1, CYC1, CYP1, ERG2, GAP1, GIT3, GLX3, GOR1, GPX2, GPX3, GRP2, HCH1, HSF1, HSP104, HSP21, HSP60, HSP70, HSP78, HSP90, IFA14, ITR1, KAR2, MDJ1, MGE1, MIA40, MNN24, MSI3, PDC11, PGA56, PGA62, POR1, ROB1, RPM2, RPN4, RPS27A, SBA1, SBP1, SGT2, SIS1, SOK1, SSA2, SSC1, STI1, TRX1, TSA1, TSA1B, YDJ1, YWP1, ZCF35, ZPR1, ZWF1

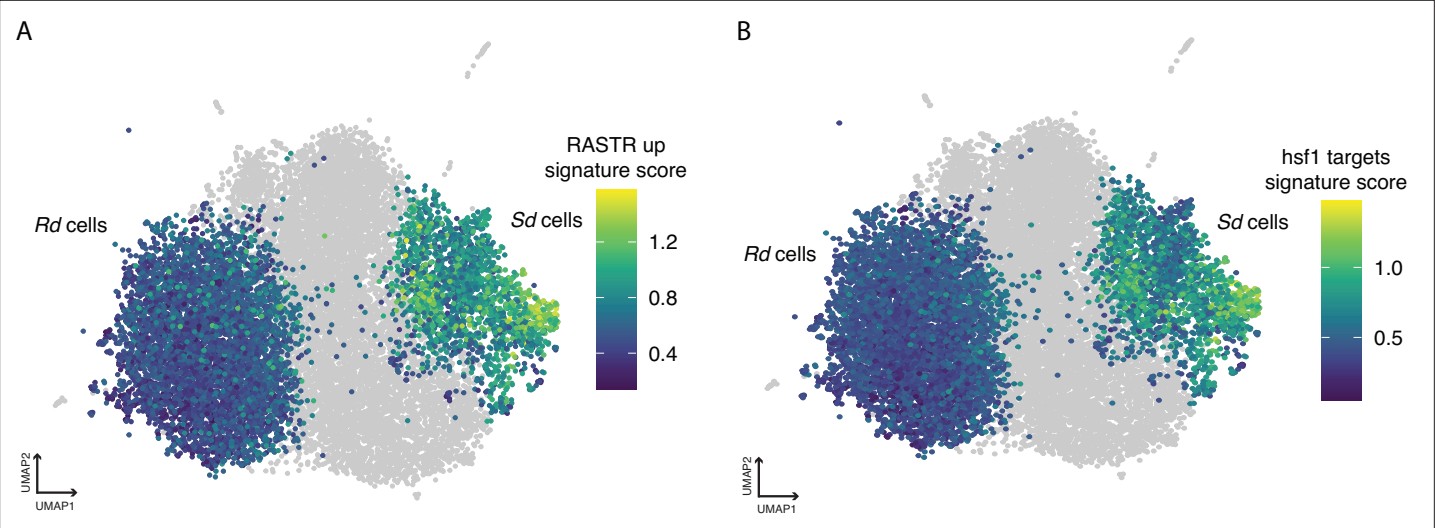

**Figure 4.** The Uniform Manifold Approximation and Projection (UMAP) embedding from *Figure 3A*, which contains fluconazole (FCZ) day 2 and day 3 cells. Here, each Ribo-dominant (*Rd*) and Stress-dominant (*Sd*) cell is colored by its signature score for: (**A**) *C. albicans* orthologs of genes upregulated in ribosome assembly stress response (RASTR) and (**B**) constitutive targets of Hsf1.

the *Rd* and *Sd* states may contribute to a RASTR-like response in the days following FCZ exposure, and based on the RASTR response, that cells in the *Rd* state may transition to an *Sd* state.

Ifh1 is also a key component of RASTR control and signaling (*Albert et al., 2019*), and the vast majority of the targets for this transcription factor are RP genes (N=144 genes of which 41 are putative or uncharacterized genes; 64 RP) (*Wade et al., 2004*). Of these, 103 are detected at significant levels in our sc profiles. As shown previously, the RP genes are highest in *Rd* and lowly expressed in *Sd*.

## The *Rd* and *Sd* subpopulations persist at 6 days post-FCZ treatment

Next, we explored whether the *Rd* and *Sd* subpopulations would persist once they were no longer exposed to the drug. On day 3 the FCZ-exposed cultures were transferred to YPD without drug and sc profiling was conducted 3 days later (FCZ day 6, *Figure 1A*, Materials and methods, 'Strains, media, and drug treatment'). *Figure 5A* shows the UMAP embedding of the sc transcriptional profiles, using only the FCZ day 3 and FCZ day 6 survivor populations.

Leiden clustering was used to identify clusters of similarly behaving cells. Cluster stability analysis however did not find strong support separating clusters 1 and 2 (panel B), meaning that they could be collapsed together. FCZ day 3 cells are primarily located on the right in clusters 2 and 4; FCZ day 6 cells are primarily located on the left in clusters 1 and 3 (panels C and F). We detected the *Rd* response, predominantly in clusters 1 and 2. Specifically, these two clusters have RP and low HSP scores (*Figure 5D and E*). The *Sd* response is predominantly in clusters 3 and 4. Note that both the *Rd* and *Sd* subpopulation are detected in FCZ day 6 cells (cluster 1 and cluster 3, respectively). This suggests that the bifurcation is still present 3 days post-drug treatment after receiving fresh growth media.

Furthermore, because clusters 1 and 2 represent cells with only minor differences in the underlying expression patterns (because the two clusters are not stable), and because day 3 FCZ cells are primarily in cluster 2 and day 6 FCZ cells are primarily in cluster 1, we can conclude that day 3 and day 6 FCZ *Rd* cells have strong similarities. Given the strong resemblance in expression patterns between the *Rd* cells at day 3 and day 6, it is likely that the day 6 cells are 'descendants' of day 3 cells. Moreover, these Rd cells largely remain in the same state regardless of the increased time since FCZ exposure, suggesting Rd cells remain viable or largely unchanging at the transcriptional level.

Clusters 3 and 4 harbor day 6 FCZ and day 3 FCZ *Sd* cells, respectively. There are significantly differentially expressed genes (DEGs) between these two groups for example, the alcohol dehydrogenase signature is most highly expressed at day 6 (*Figure 5G and H*). Heat-shock stress is slightly reduced in day 6 compared to day 3 (*Figure 5G*). We also observe a substantial change in the relative proportions of the *Rd* and *Sd* populations: at day 3 we observe 70% *Rd* and 30% *Sd* but at day 6 we

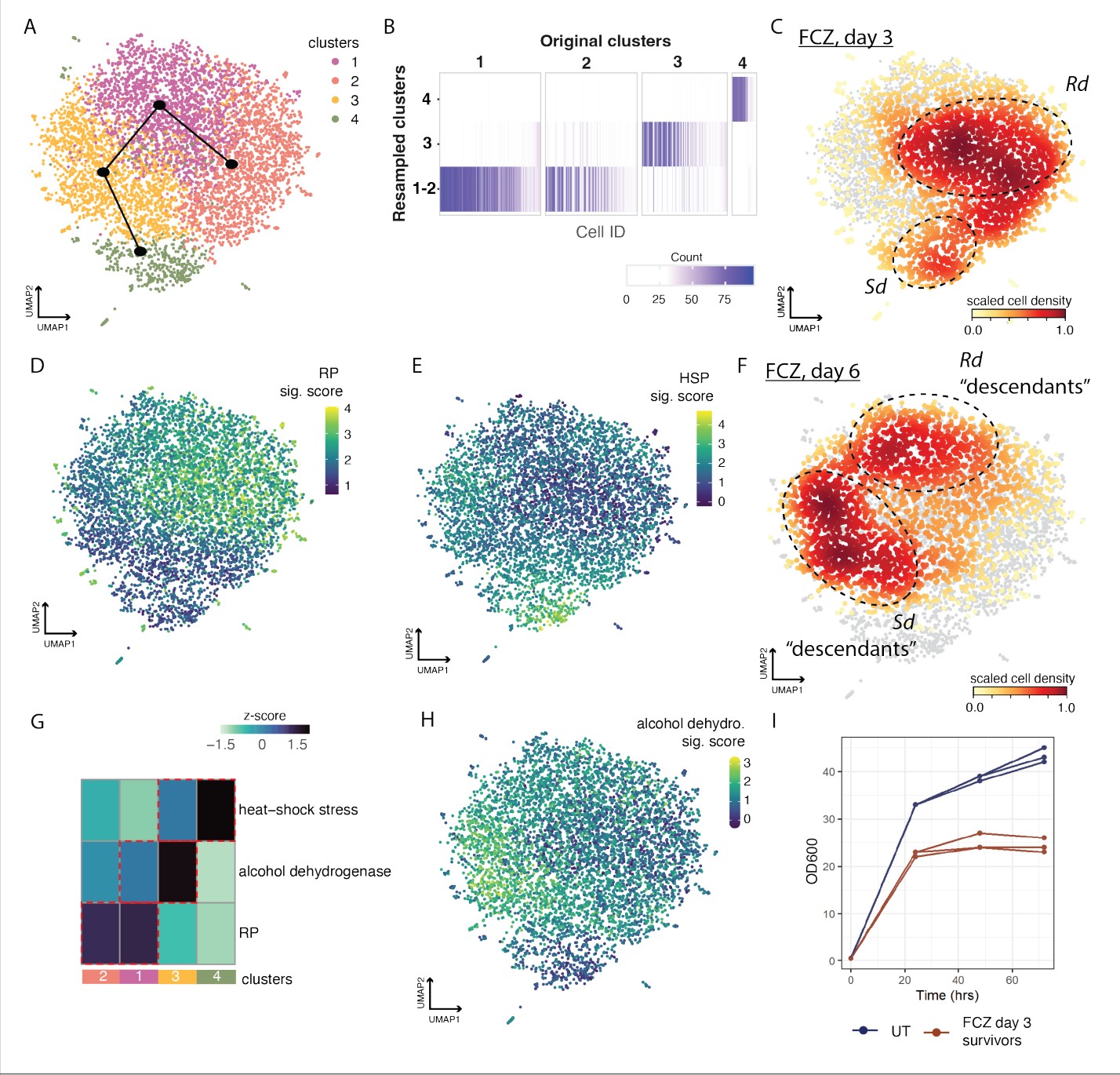

**Figure 5.** Both the Rd and Sd states persist at six days post-fluconazol treatment. (**A**) Uniform Manifold Approximation and Projection (UMAP) embedding of fluconazole (FCZ)-treated cells at days 3 and 6 (after resuspension in fresh media at day 3). Leiden clustering identified four clusters. Clusters are related along the trajectory (black) from slingshot analysis. The topology suggests that FCZ Ribo-dominant (*Rd*) cells at day 6 evolve from FCZ *Rd* cells at day 3. This relationship is also true for FCZ Stress-dominant (*Sd*) cells. (**B**) The heatmap depicts cluster stability analysis for the four clusters. Purple ticks depict the number of times each cell was assigned to each of the four clusters. Cluster stability analysis failed to strongly separate clusters 1 and 2. However, both of these clusters correspond to *Rd* cells. (**C–F**, **H**) The UMAP embedding of cells from (**A**) was annotated with color bars in C–F, H to depict: (**C**, **F**) density of cells from day 3 in FCZ and (**F**) from day 6 (after 3 days in YPD alone); or (**D**, **E**, **H**) signature scores for the (**D**) ribosomal protein (RP), (**E**) heat-shock stress, and (**H**) alcohol dehydrogenase gene signatures. (**G**) A heatmap depicting the average level of activation (VISION z-scores, color bar) of different signatures across clusters presented in (**A**), analogous in methodology to *Figure 2C*. (**I**) Growth curves for untreated (UT) and FCZ day 3 survivor resuspended in fresh media as measured by OD_{600} across days.

observe an almost even ratio (56% *Rd* and 44% *Sd*) (*Figure 5C* versus *Figure 5F*). Upon resuspension in fresh media, $OD_{600}$ measurements confirmed that cells exposed to FCZ for 3 days and then resuspended in fresh media and allowed to grow for 3 days without drug, proliferate more slowly, and reach lower biomass levels than UT cells propagated without drug for 3 days (*Figure 5I*). Together this suggests that *Sd* cells, unlike *Rd* cells, are significantly changing over time since the FCZ exposure (although they do retain many of the molecular properties of day 3 cells) and may have either a higher proliferative capacity or survival rate relative to Rd cells.

We also used computational trajectory analysis to investigate the relationship between these clusters (*Figure 5A*, black tree; Materials and methods, 'Cell clustering, trajectory, and signature analyses', slingshot analysis). Slingshot analysis partially orders clusters of cells based on their transcriptional profiles; the resultant order is suggestive of how cells evolve their expression patterns over time in a manner analogous to an animation. Overall, these findings suggest that there is more evidence that day 6 FCZ *Sd* cells are 'descendants' of day 3 FCZ *Sd* cells compared to, for example, day 6 Sd cells 'descending' from day 3 *Rd* cells.

## Discussion

Sc approaches can facilitate the characterization of phenotypic heterogeneity in microbial populations by enabling the detailed examination of individual microbial cells, revealing the full range of diversity in their gene expression. Here, we characterized two distinct subpopulations of cells with different expression profiles – the *Rd* and *Sd* states – among isogenic cells treated with FCZ, the most widely used antifungal drug. This result supports the observation that antifungal tolerance is due to a subpopulation of cells that grows in the presence of FCZ, while other cells in the population do not.

Despite the technical challenges associated with sc profiling of yeast cells, we identified heterogeneity in the gene expression of UT prototrophic SC5314 *C. albicans* cells that reflects the cell cycle phase and associated metabolic and stress responses. Specifically, glycolysis and several heat-shock genes were more highly expressed during M phase, and oxidative stress genes were more highly expressed in a distinct subset of cells that appear to be from G1/S phase. The existence of these distinct subpopulations is consistent with observations in the literature (*Chiu et al., 2011*; *Finkel and Hwang, 2009*; *Hossain et al., 2021*; *Senn et al., 2012*). The magnitude of stress signatures is much lower than in cells exposed to antifungal drugs. We also validated the distinct subpopulations by fluorescence microscopy with tagged genes differentially expressed in different cell cycle stages. Thus, the sc transcriptional profiling detects cell-to-cell heterogeneity within UT *C. albicans* cultures.

We then examined the transcriptional profiles of individual drug-treated cells. During FCZ exposure, distinct subpopulations of *C. albicans* appeared: on day 2, FCZ-treated cells were either in the *Rd* or *Sd* state with a higher frequency of Rd cells. By day 3, the *Sd* subpopulations become more balanced suggesting either the *Sd* subpopulation is more proliferative, or perhaps that cooperation between the two subpopulations is required, for example, for the exchange of metabolites. Before and after treatment, the vast majority of cells are in the white morphology so *C. albicans* morphological state does not explain this *Rd/Sd* population split.

The *Rd* response is characterized by high expression of ribosomal protein (RP) and ribosomal RNA (rRNA) processing genes. An imbalance of RP and rRNA can lead to proteotoxic stress and RP aggregation in *S. cerevisiae* (*Tye et al., 2019*), and might do so as well in *C. albicans*. The *Sd* state involves the heat-shock transcription factor Hsf1, which upregulates *HSP90*, *HSP70*, and other genes in the UPR, which have been shown to suppress RP expression in *S. cerevisiae* to protect against toxic protein build-up, and restores cell growth (*Albert et al., 2019*). The molecular profile of *Sd* cells is consistent with this transition exhibiting high expression of RASTR-induced genes, for example, Hsf1 constitutive target genes including *HSP70*, *HSP90*, and disaggregases *HSP21*, *-60*, *-78*, and *-104*, along with low expression of RP genes and rRNAs. Importantly, *Sd* cells also highly express several genes associated with drug tolerance (*Delarze et al., 2020*; *Garnaud et al., 2018*; *Liu et al., 2015*; *Mayer et al., 2013*; *Nagao et al., 2012*; *Rosenberg et al., 2018*) as well as genes involved in filamentous growth, adhesion, and biofilm formation. This is consistent with a potential role of Hsf1 (*Hahn et al., 2004*; *Leach et al., 2016*; *Leach et al., 2012*; *Veri et al., 2018*) as a trigger of RASTR.

Overall, the molecular profiles of the *Rd* and *Sd* states suggest a model in which some fraction of the cells exposed to FCZ initially enter the *Rd* response and trigger the RASTR response (*Albert et al., 2019*) which in turn causes the cells to transit into the *Sd* state. This could suggest that *Sd* cells may

represent a *C. albicans* subpopulation that have mounted a successful drug tolerance stress response. Alternatively, it could be the case that both subpopulations are necessary for cell survival.

We also treated cells with RAPA (analysis in Appendix 2). RAPA in isolation is considered to have weak antifungal properties, but drug tolerance (but not drug resistance) is ablated when used in combination with FCZ (*Rosenberg et al., 2018*). We did not identify any RAPA-treated cells in the Rd state. This is perhaps not surprising given that RAPA targets Tor1 and shuts down ribosomal biogenesis, an important component of the Rd state. In fact, RAPA-exposed cells also had the weakest RP expression. Of note, RAPA specifically triggers the expression of genes involved in hyperosmotic stress. Previous work established that the Tor1 pathway controls phase separation and the biophysical properties of the cytoplasm by tuning macromolecular crowding (*Delarue et al., 2018*). Hyperosmotic stress response to RAPA exposure might reflect disruption of the biophysical properties of the cytoplasm, altering the osmotic balance in cells.

Whereas nearly every RAPA-treated cell had an *Sd* transcriptional profile, nearly every CSP-treated cell after 2 days exposure had an *Rd*-like transcriptional profile (analysis in Appendix 2). The echinocandin CSP is fungicidal with a mechanism of action distinct from FCZ – it inhibits β-1,3 glucan synthesis and disrupts the fungal cell wall. We did not find any CSP-treated cells in the *Sd* subpopulation by day 3. Although we used low concentrations of CSP to avoid cell aggregation and therefore it is possible that alternative CSP concentrations could induce *Sd* cells, it appears that survivors of the fungicidal drug are transcriptionally far more homogeneous at early time points (days 2–3).

After resuspension in fresh media and 3 days of growth (day 6), both the *Sd* and *Rd* subpopulations persist. In fact, now *Sd* outnumbers the *Rd* subpopulation. Their persistence hints that these surviving cells have some 'memory' of treatment with FCZ. In the case of *Rd* cells, there are few transcriptional changes between days 3 and 6. For *Sd*, there is a downregulation of the heat-shock response and an increase in the alcohol dehydrogenase pathway, suggestive of cellular proliferation in the fresh medium. *Sd* cells at day 6 still however largely retain differential expression of genes, pathways, and proteins identified in *Sd* cells at day 3.

## Conclusions

The use of a nanoliter droplet-based assay adapted for fungal cells enabled a detailed sc analysis of *C. albicans* in the absence and presence of antifungal drugs over a period of days. The assay was cost-effective and encountered minimal issues or failed runs. Studies of thousands of individual cells enriched our understanding of community structure and population heterogeneity. Specifically, this study refines published bulk transcriptome studies by differentiating between genes, pathways, and responses that are expressed universally in all cells, and those that are restricted to specific subpopulations. Here, by examining cellular trajectories across populations in a high-throughput manner, we obtained new insights into time-sensitive processes, such as the emergence of drug tolerance during 2–3 days of drug exposure and the existence of cells with *Rd* and *Sd* states, which reflect two different cellular states within the RASTR response described as a molecular signature of an *S. cerevisiae* stress response. If the *Rd* response proves generalizable to other drug treatments, disrupting the *Rd* to *Sd* tolerance transition or the *Sd* response itself could represent an innovative therapeutic approach for other antifungal treatments.

## Materials and methods
### Strains, media, and drug treatment
#### *C. albicans* cultures for sc-RNA-, bulk RNA- and DNA-sequencing

*C. albicans* SC5314 cells were streaked out from glycerol stocks in –80°C on YPD agar plates (2% D-glucose, 2% peptone, 1% yeast extract, 0.01% uridine, 2% agar) and incubated at 30°C for 48 hr. Afterward, a single colony of cells was transferred into YPD liquid media (2% D-glucose, 2% peptone, 1% yeast extract, 0.01% uridine) and incubated at 30°C for 12–16 hr.

#### Preparation for sc profiling via DROP-seq

For UT cells, a single colony of SC5314 cells were transferred to 5 ml YPD and grown overnight to yield ~$10^8$ cells/ml. Cultures were then diluted to an $OD_{600}$ of 0.1 in fresh 50 ml YPD liquid and incubated at 30°C in a shaker incubator. Cells were collected when OD reached 0.5–0.9 in order to

maximize the number of cells in mid-log phase. Cells were pelleted by centrifugation, 1 ml of RNAlater was added (Sigma # R0901), the suspended cells were incubated for 10 min at room temperature, and the resulting culture was frozen at –20°C for later use.

For treated cells, we performed the following approach to ensure a sufficient number of survivor cells for profiling across different drugs and time points: cells were pelleted and resuspended in 1 ml of YPD, 250 µl of this suspension was combined with 15 ml of fresh YPD and incubated at 30°C with shaking until the culture reached an $OD_{600}$ of 0.5–0.6. Finally, ~$10^8$ of these cells were seeded into 10 ml of YPD. Each suspension was then subjected to drug treatment.

FCZ (Sigma #F8929) was used at 1 µg/ml, which is from 1× to 2× the dosage relative to the $MIC_{50}$ for SC5314 in YPD (*Figure 3—figure supplement 1*). CSP (Sigma #SML0425), a compound that interrupts cell wall biosynthesis (*Cappelletty and Eiselstein-McKitrick, 2007*; *Stevens et al., 2004*; *Yang et al., 2017*), was used at 1 ng/ml, which is well below the reported $MIC_{50}$ and was chosen to ensure a sufficient number of non-aggregated survivors to generate sc profiles. For RAPA, a subinhibitory concentration of 0.5 ng/ml was chosen based on previous studies that established such levels are sufficient to generate a fungicidal synergistic effect when given concomitantly with FCZ (*Rosenberg et al., 2018*). Each drug was added to the individual cultures and incubated at 30°C for 48 or 72 hr. For the day 6 population, day 3 survivors were washed twice, and resuspended in 10 ml of fresh YPD.

At each time point, cultures were collected and strained (pluriStrainer 20 µm, pluriSelect) before placement in fresh tubes. Straining was done in order to minimize the likelihood of clogging in the microfluidic due to rare but large hyphae and pseudohyphae morphologies. We observed that germ tubes up to four times the length of the mother cell can still be processed for DROP-seq analysis. Such cells are well within the hyphal transcriptional profile (*Nantel et al., 2002*), suggesting that our results may contain some profiles of hyphae and pseudohyphae cells. After filtering, the vast majority of cells were in the yeast white morphology with less than 0.2% of cells in a filamentous morphology (hyphae or pseudohyphae) after manual counting ~100 microscopy images with an average of ~50 cells per slide for each such population. All cultures yielded a sufficient population of survivors for downstream sc transcriptional profiling, bulk transcriptional profiling, bulk DNA genomic profiling, and/or microscopy. Cultures were washed with 1 ml of RNAlater twice. Cells were then resuspended in 1 ml RNAlater and incubated at room temperature for 10 min before storage at –20°C until sc profiling with DROP-seq.

## Cultures for $OD_{600}$ analyses

Three cultures were inoculated with single cultures and incubated overnight in YPD at 30°C. The following morning, cultures were diluted to an $OD_{600}$ of ~0.5 in either fresh YPD or YPD with 1 µg/ml FCZ (same concentration used for the DROP-seq experiment). $OD_{600}$ was measured immediately without dilution, then every 24 hr via a 1:100 dilution for a total of 72 hr (*Figure 3—figure supplement 1A*). To investigate the growth of day 3 FCZ survivors, three cultures previously grown for 72 hr in either YPD or YPD with 1 ug/ml FCZ were diluted to an $OD_{600}$ of ~0.5 in fresh YPD without drug. $OD_{600}$ was measured immediately without dilution and then every 24 hr via a 1:100 dilution for a total of 72 hr (*Figure 5I*).

## Spheroplasts

The *C. albicans* setting required an optimized protocol for the removal of the cell wall and to induce stable spheroplasts for sc profiling. Toward this end, we experimented with different concentrations of zymolyase (0.1, 0.2, and 0.4 U zymolyase (BioShop # ZYM002) with $10^7$ cells in 100 µl of sorbitol 1M) at different time points (incubated at 37°C for 10, 20, 30 min) before processing with the DROP-seq. To compare cultures grown under different conditions, cells were stained with calcofluor white and imaged using a Leica DM6000 microscope. We concluded that concentrations in the range 0.15–0.25 U after 25 min are able to induce spheroplasts that remain sufficiently stable for processing with our DROP-seq.

## Sample preparation for sc profiling

At the time of DROP-seq profiling, an aliquot of $10^7$ ($OD_{600}$=0.68) cells was separated from the culture in Materials and methods, 'Strains, media, and drug treatment', and washed three times with sorbitol 1M solution. The cells were then resuspended in 100 µl sorbitol 1M+0.25 U zymolyase and incubated

at 37°C for 25 min (as per our findings in Materials and methods, 'Spheroplasts'). Next, the cells were pelleted and resuspended again in 0.5 ml of cold and fresh RNAlater for 5 min. Now, the cells were washed (centrifuged and pelleted) with 1 ml of washing buffer (1 M sorbitol, 10 mM TRIS pH 8, 100 µg/ml BSA) three times. Finally, $10^6$ cells (OD$_{600}$=0.08) were resuspended in 1.2 ml of the washing buffer. This cell suspension was then used as input to the DROP-seq device.

Sample preparation generally follows the protocol given by *Macosko et al., 2015*, with some exceptions. Whereas Macosko et al. recommend a ratio of 100K mammalian cells to 120K beads for DROP-seq, we found that a ratio of 1M cells for 120K beads generated a sufficient yield of cDNA as per the Tapestation (Agilent Inc) device. *Jackson et al., 2020*, use 5M cells as input to the Chromium (10X Inc) system. Furthermore, whereas Macosko et al. use 1 ml of lysis buffer, we use 1.2 ml. Instead of 13 PCR cycles, we use 17 (Jackson et al. uses 10 cycles). Samples were sequenced using a NEXT-seq 500 (Illumina Inc) following the standard Macosko et al. protocol set to yield an estimated 200 million reads/sample.

## Quality control, basic processing, and normalization of the sc profiles

All computations were performed using `Python` version 3.9.6 (*van Rossum and Drake, 2009*) or R version 4.0.4 (*R Development Core Team, 2021*). Gene abundances were estimated from raw sequencing data using the end-to-end pipeline `alevin-fry` (*He et al., 2021*) which performs UMI deduplication and reduces the number of discarded (multimapped) reads. The pipeline utilizes a reference index covering the spliced transcriptome extracted from the latest version of *C. albicans* strain SC5314_A22 (haplotype A, version 22; GCF_000182965.3). The Unspliced-Spliced-Ambiguous (USA) mode was used to separately keep track of the types of transcripts from which UMIs are sampled. A gene-by-cell matrix for each sample was obtained by summing reads labeled as either 'spliced' or 'ambiguous' by `alevin-fry`.

We started by filtering genes from downstream analyses with a zero sum count (the count across all cells), as were cells with less than 5, or more than 2000 transcript counts (*Figure 1—figure supplement 1A*). Droplets almost always capture ambient 'free floating' RNA that is present in the suspension. Therefore, even when a microparticle is not captured alongside a cell in a droplet, sequencing still yields reads originating from this ambient RNA. However, the number of such reads is far lower than a droplet with a successful cell capture. Cells were included in their analysis if their profiles significantly deviated from levels indicative of ambient RNA in the suspension using EmptyDrops (FDR <0.01) (*Lun et al., 2018*). SCANPY (*Wolf et al., 2018*), a python-based toolkit, and SingleCell-Experiment (*Amezquita et al., 2020*), a R/Bioconductor package, were used for data quality control, filtering of genes and downstream visualization. We removed genes (n=869) that were expressed in less than 20 cells (*Figure 1—figure supplement 1C*).

We then use scVI (*Gayoso et al., 2021*; *Lopez et al., 2018*) version 0.12.2, a Bayesian deep neural network architecture which implements a probabilistic model of mRNA capture and uses a variational autoencoder to estimate priors across batches and conditions. Models were built using default parameters for different grouping of samples: (i) UT cells, (ii) UT cells and CSP, RAPA, FCZ treated at day 2 (and 3), and (iii) FCZ-treated cells at days 3 and 6. All models were adjusted for batch and library size. We trained `scVI's` variational autoencoder and stored the latent representation for visualization and downstream analyses. We reduced the inferred latent spaces to two dimensions via the UMAP tool using the implementation of `umap-learn` (*McInnes et al., 2020*) in `SCANPY` (*Wolf et al., 2018*) (min_dist = 0.3). When analyzing the level of expression of individual genes, missing or dropout values were first imputed using MAGIC (*van Dijk et al., 2018*).

## Bulk transcriptomics

Total RNA was extracted from FCZ-treated cells at day 2 post-exposure, which were grown according to Materials and methods, 'Preparation for sc profiling via DROP-seq:', using the QIAGEN RNeasy mini kit protocol. RNA quality and quantity were determined using a Bioanalyzer (Agilent Inc). Paired-end read sequencing (2×50 bp) was carried out on a NextSeq500 sequencer (0.5 Flowcell High Output; Illumina Inc). Raw reads were pre-processed with the sequence-grooming tool `cutadapt` (*Martin, 2011*) version 0.4.1 with quality trimming and filtering parameters: `--phred33 --length 36 --2colour 20 --stringency 1 -e 0.1`. Each read pair was mapped against *C. albicans* strain SC5314_A22 (haplotype A, version 22; GCF_000182965.3) downloaded from the NCBI using `STAR` (*Dobin et al.,*

*2013*) version 2.7.9a with the following filtering parameters: `--outSAMmultNmax 1 --outSA-Munmapped Within --outSAMstrandField intronMotif`. We obtained ~13 million reads of which 88% were uniquely mapped along the genome. The read alignments and *C. albicans* genome annotation strain SC5314_A22 (haplotype A, version 22; GCF_000182965.3) were provided as input to `featureCounts()` from the Rsubread package (*Liao et al., 2019*) version 2.4.3 to estimate gene abundances. The following parameters were used: `isPairedEnd = TRUE, countReadPairs = TRUE, requireBothEndsMapped = TRUE, checkFragLength = FALSE, countChimeric-Fragments = FALSE, countMultiMappingReads = TRUE, fraction = TRUE`.

## Construction of pseudo-bulk profiles

Throughout the manuscript, pseudo-bulk profiles refer to transcriptional profiles that are derived from the sc reads by ignoring barcodes. This pipeline is depicted in *Figure 1—figure supplement 2A*. By ignoring the R1 (left) read of the sc profile that contains the cellular barcode, we are effectively performing 'bulk' RNA-sequencing using only the R2 (right) read that aligns to a transcript in the sample. This collapses all cells to a single profile. We compared two different techniques to compute pseudo-bulk profiles, or we can first use the barcoded reads to partition cells into classes (e.g., *Rd* versus *Sd*) and then form a pseudo-bulk profile specific to each of the classes. The *unfiltered* pseudo-bulk data is derived from counting raw reads aligned to the reference genome using the `STAR` tool (*Dobin et al., 2013*). The *filtered* pseudo-bulk dataset is obtained by first applying our sc pipeline (`alevin-fry` followed by `EmptyDrops`) and then summing across all cells. The first is closer in spirit to true bulk (single read) profiling, while the second approach filters reads, cells, and genes in the same manner as sc analyses and therefore represents a middle point between bulk and sc profiling. A comparison of *filtered* pseudo-bulk FCZ profiles at day 2 and 3 datasets indicated that the assay is robustly quantifying the expression of genes across different batches (*Figure 1—figure supplement 2A*).

A comparison of bulk profiles versus *unfiltered* pseudo-bulk for the FCZ profiles at days 2 and 3 indicated that both methods identified all but 297 of the same genes. The missed genes tended to be expressed at low levels in the bulk profiles (*Figure 1—figure supplement 2B*). Moreover, day 2 and day 3 pseudo-bulk profiles were significantly correlated with 'true' bulk RNA-sequencing profiles (R ranges from 0.67 to 0.74; *Figure 1—figure supplement 2C*). Here 'true' bulk RNA-sequencing profiles were generated as described in Materials and methods, 'Bulk transcriptomics'.

## Whole-genome DNA-sequencing

*C. albicans* populations were grown as described in Materials and methods, 'Strains, media, and drug treatment', although we did not apply a cell filtration step to remove filamentous cells. Preparation of genomic DNA used the MasterPure Yeast DNA Purification Kit (Lucigen # MPY80200) with a NextSeq500 – 1 flowcell mid output (130M fragments), 150 cycles pair-end reads (maximum 2×80 nt), yielding on average 23.9 million reads per sample (1 UT, FCZ at days 2, 3, 6, and 12). This gives an expected sequencing depth of 224 since the size of the *C. albicans* genome is ~16 Mb.

Raw reads were pre-processed with the sequence-grooming tool cutadapt (*Martin, 2011*) version 0.4.1 with quality trimming and filtering parameters: `--phred33 --length 36 -q 5 --stringency 1 -e 0.1`. Analyses followed a previous study by *Ford et al., 2015*, which also examined *C. albicans* complete genomes, although we used a more recent version A22 of the *C. albicans* SC5314 haplotype A genome for read mapping (downloaded from the Candida Genome Database; https://www.candidagenome.org/). Briefly, reads were mapped using the BWA alignment tool (*Li and Durbin, 2009*) and `RealignerTargetCreator` and `IndelRealigner` from GATK (*McKenna et al., 2010*) were used for re-alignment. SNPs were detecting using Unified Genotyper (GATK version 1.4.14) using the same filtration criteria as reported in Ford et al. Determination of copy number and loss-of-heterozygosity was performed using GATK with the strategy reported in Ford et al.

Restricting attention to the UT samples, our observed sequencing depth was just under 200 and we identified approximately the same number of single nucleotide polymorphisms (N=3304) and the same number of insertions/deletions (181/255 resp.) as a previous whole-genome sequencing effort (*Cuomo et al., 2019*) using their bioinformatic pipeline. Although mutations were observed in some reads at some genomic loci, the samples were not enrichment for mutations that occurred more often than the rate of sequencing error which is $10^{-3}$ after correcting for multiple testing. This error rate is

in line with estimates of the spontaneous mutation rate for *C. albicans* (*Ene et al., 2018*), together suggesting that the population is near isogenic without any significantly large subclones. Given that the error rate for copy number variants (e.g. loss, amplification) is lower than polymorphisms (*Ene et al., 2018*) and the duration of cell expansion before drug exposure (<2 days), it would be unlikely that spontaneous mutations explain the degree of heterogeneity that was observed 2–3 days post-drug exposure.

## Cell clustering, trajectory, and signature analyses

To identify subpopulations of cells with similar gene expression patterns in an unbiased, unsupervised manner, we applied Leiden clustering (*Traag et al., 2019*) on the latent space generated by scVI (resolution of 0.5 for the analysis combining UT, FCZ day 2 and 3, CSP, and RAPA-treated cells and resolution of 0.4 for FCZ day 3–6 analyses). To test the robustness of the clustering process, we repeated the clustering process 100 times, each time using a random subset of 95% of the cells. With each such random set, we repeated the scVI model building, followed by Leiden clustering to identify clusters. The mapping from the original clusters to the new clusters was then established based on the maximum overlap in cell membership between the original and newly formed clusters. Finally, we counted the frequency that each cell was assigned to each cluster over all the iterations.

The lineage trajectory of day 3 and 6 FCZ-treated cells was performed using the R/Bioconductor `slingshot` package version 1.8.0 with default parameters (*Street et al., 2018*).

To investigate the key sources of variability across cell subpopulations, we curated 43 gene signatures related to microbial phenotypic diversity and drug tolerance including the cell cycle, TCA cycle, specific and general stress responses, metabolic pathways, amino acid biosynthesis, efflux pumps, and specific drug responses among others (*Azadmanesh et al., 2017*; *Berman, 2006*; *Côté et al., 2009*; *Enjalbert et al., 2006*; *Enjalbert et al., 2003*; *Gasch et al., 2000*; *Hardwick et al., 1999*; *Hinnebusch, 2005*; *Jackson et al., 2020*; *Natarajan et al., 2001*; *O'Duibhir et al., 2014*; *Pais et al., 2019*; *Sanglard, 2016*; *Sanglard et al., 2003*; *Tsai et al., 2019*; *Yang et al., 2017*; *Figure 2—source data 1*). In some cases, gene signatures from the literature were derived in other organisms and required orthology mappings to *C. albicans*.

Briefly, signatures of *cell cycle phases* were identified as transcriptional expression patterns in synchronous *C. albicans* populations (*Berman, 2006*; *Côté et al., 2009*) or were expert-curated and well-established cell cycle genes found in distinct clusters of *S. cerevisiae* sc profiles (*Jackson et al., 2020*). Signatures *specific to certain stresses* were found by transcriptional profiling of *C. albicans* challenged by temperature, osmotic and oxidative stress under conditions that permitted >60% cell survival (*Enjalbert et al., 2003*) or in *C. glabrata* (*Pais et al., 2019*). We also curated more *general non-specific stress signatures* identified in *C. albicans* (*Enjalbert et al., 2006*) or *S. cerevisiae* (*Gasch, 2007*; *Gasch et al., 2017*; *Tsai et al., 2019*). Previous studies established the existence of a ubiquitous environmental stress response (ESR) in *S. cerevisiae* which is deregulated in response to many different environmental perturbations (*Gasch, 2007*; *Gasch et al., 2000*). The ESR is divided into the induced (iESR) and repressed (rESR) subcomponents. The iESR is characterized by overexpression of heat-shock and oxidative stress genes in addition to genes involved in central carbohydrate metabolism and energy generation (*Gasch, 2007*; *Gasch et al., 2000*). Previous studies have noted a complex, intricate relationship between the ESR and cell cycle phase (*O'Duibhir et al., 2014*; *Regenberg et al., 2006*). Some investigations suggested a smaller core ESR in *C. albicans* (*Enjalbert et al., 2006*) which may have evolved due to the unique host environment with the need to grow with different substrates (*Brown et al., 2014*). Curated *metabolic signatures* include lowly expressed glycolytic genes and highly expressed TCA cycle genes identified during diauxic shift or RAPA treatment in yeast (*DeRisi et al., 1997*; *Hardwick et al., 1999*) as well as GCN4-driven amino acid biosynthesis, another well-defined metabolic signature conserved between *S. cerevisiae* and *C. albicans* (*Hinnebusch, 2005*; *Natarajan et al., 2001*).

The signature analyses start with the VISION tool which estimates a signature score for every cell (*DeTomaso et al., 2019*) using the batch-adjusted normalized counts returned by scVI'smodel. The distribution of individual scores for cells classified in each cluster can be depicted using the empirical cumulative distribution function. These distributions were further compared using the Kolmogorov-Smirnov test. We then used the median to summarize signatures scores of cells within each cluster and selected signatures which were the most variable across clusters (sd >0.05 for the analysis combining

UT, FCZ- day 2 and 3, CSP-, and RAPA-treated cells or combining FCZ-treated cells at days 3 and 6; ). Heatmaps were used to depict the median scores (z-score, color bar) of the selected signatures for each cluster.

## DGE analysis

### Sc differential gene expression (DGE)

`scVI`'s model allows us to approximate the posterior probability of the alternative hypotheses (genes are different) and that of the null hypotheses through repeated sampling from the variational distribution, thus obtaining a low variance estimate of their ratio (i.e., Bayes factor). We used this approach to identify genes differentially expressed in the combined and individual comet clusters compared to the other clusters (*Appendix 2—figure 2A, B*).

### Pseudo-bulk DGE and gene ontology (GO) enrichment analysis

We also conducted pseudo-bulk differential gene expression analyses which allow for a dramatic reduction in the number of zeros in the data by aggregating cells within each replicate. This approach has been found to achieve the highest fidelity to the experimental ground truths significantly reducing the risk of false discoveries (*Squair et al., 2021*). As described in Materials and methods, 'Construction of pseudo-bulk profiles', *filtered* pseudo-bulk profiles were obtained by summing counts of selected cells within each replicate. In order to use `DESeq2` (*Love et al., 2014*), a standard R/Bioconductor package used for differential analysis of count data, we require at least two replicates within each group of comparison. If only one replicate was available (e.g., FCZ at day 6), we partitioned the selected cells of a single replicate into two groups randomly before forming pseudo-bulk profiles. We then used `DESeq2` default parameters to perform DE analysis and selected genes with Benjamini-Hochberg FDR <0.1 (*Benjamini and Hochberg, 1995*). Models were adjusted for batch affects where relevant.

Finally, we identified enrichment of biological processes in lists of significantly over- and under-expressed genes using the R/Bioconductor `ViSEAGO` package (*Brionne et al., 2019*). `ViSEAGO` includes all algorithms developed in the R/Bioconductor `topGO` package including the weight01 fisher test that takes into account the topology of the GO graph (*Alexa et al., 2006*). Biological processes with weight01 p-value <0.01 were defined as significantly enriched in the gene list.

## Cell imaging

To validate the subpopulation structure identified by the sc transcriptomics, we choose markers representative of distinct clusters. Cells were transformed with *GFP* and *RFP* fusion constructs for *HSP70* and *TTR1* marker genes respectively using a CRISPR/Cas9 protocol (*Min et al., 2016*) with primers described in *Figure 2—source data 2*. Strain SN76(*his1Δ/his1Δ, arg4Δ/arg4Δ, ura3Δ/ura3Δ*) was chosen for gene tagging since it is a derivative strain of SC5314 but with multiple auxotrophic markers. These markers allow for convenient selection of successfully transformed cells.

Benchling (https://benchling.com) was used to design the sgRNAs. We followed the CRISPR/Cas9 protocol with the plasmid pV1093 from *Min et al., 2016*. This includes two PCRs to fuse the SNR52 promoter to the sgRNA scaffold and terminator. The third PCR amplifies the final sgRNA cassettes. Two different plasmids pENO1-iRFP-NATr (Plasmid #129731, Addgene Inc) and pFA-GFP-HIS1 were used to design the repair segments. The construction of the Cas9 cassette proceeded as per Min et al. Amplification of the Cas9 cassette with PCR used the following schedule: 98°C for 3 min, 98°C for 30 s, 63°C for 30 s, 72°C for 5 min and 30 s. Steps 2–4 have been repeated for 34 rounds followed by 72°C for 10 min and finally the reaction finished in 4°C. The repair DNA must be amplified with the designed primers (*Figure 2—source data 2*) in 8–12 PCR tubes with 0.1 µl plasmid (500 ng/ml), 2.5 µl forward primer, 2.5 µl reverse primer, 1 µl 10 mM dNTP, 33.65 µl nuclease free water, 10 µl 5× HF PCR buffer, and 0.25 µl phusion polymerase in each tube.

Cells that were successfully transformed were grown and harvested in a manner identical to that used for the sc experiments (Materials and methods, 'Strains, media, and drug treatment'). At time of microscopy, cells were collected, washed with $H_2O$, and transferred to minimum media to minimize the background noise from normal YPD media. Afterward, cells were mounted onto the uSlide and imaged with a Leica DM6000 microscope at 1000× (~50 images/time point; ~50 cells/image).

## Acknowledgements

We thank L Pachter and the organizers of BIRS workshop #17w5134 (Oaxaca, Mexico) for several conversations that motivated this effort especially with respect to affordable open biotechnology. We thank CA Jackson and D Gresham for early access to their protocol, C Law for assistance with the microscopy, A Villani for several nice optimizations, and members of the McCarroll lab for their careful advice and outstanding responsiveness.

## Additional information

### Funding

| Funder | Grant reference number | Author |
| --- | --- | --- |
| Natural Sciences and Engineering Research Council of Canada | RGPIN-2018-05085 | Michael T Hallett |
| Canada Foundation for Innovation | 37083 | Michael T Hallett |
| European Research Council | 951475 | Judith Berman |

The funders had no role in study design, data collection and interpretation, or the decision to submit the work for publication.

### Author contributions

Vanessa Dumeaux, Software, Formal analysis, Investigation, Visualization, Methodology, Writing – original draft, Writing – review and editing; Samira Massahi, Formal analysis, Investigation, Methodology, Writing – review and editing; Van Bettauer, Software, Formal analysis, Investigation, Visualization, Methodology; Austin Mottola, Validation, Investigation, Writing – original draft, Writing – review and editing; Anna Dukovny, Investigation, Writing – review and editing; Sanny Singh Khurdia, Shawn Simpson, Investigation; Anna Carolina Borges Pereira Costa, Raha Parvizi Omran, Resources; Jinglin Lucy Xie, Writing – review and editing; Malcolm Whiteway, Resources, Writing – review and editing; Judith Berman, Investigation, Writing – original draft, Writing – review and editing; Michael T Hallett, Conceptualization, Supervision, Funding acquisition, Methodology, Writing – original draft, Project administration, Writing – review and editing

### Author ORCIDs

Vanessa Dumeaux  https://orcid.org/0000-0002-1280-6541
Malcolm Whiteway  http://orcid.org/0000-0001-6619-7983
Michael T Hallett  http://orcid.org/0000-0001-6738-6786

### Decision letter and Author response

Decision letter https://doi.org/10.7554/eLife.81406.sa1
Author response https://doi.org/10.7554/eLife.81406.sa2

## Additional files

### Supplementary files

• MDAR checklist

### Data availability

Python/R code and data required for reproducibility is available through the Open Science Foundation (OSF) repository https://osf.io/5tpk3/ and associated github repository https://github.com/vdumeaux/sc-candida_paper (copy archived at *Dumeaux, 2023*). The raw and processed single-cell transcriptome and bulk RNA-seq is also available through NCBI's Gene Expression Omnibus with accession number GSE204903.

The following datasets were generated:

| Author(s) | Year | Dataset title | Dataset URL | Database and Identifier |
|---|---|---|---|---|
| Dumeaux V, Massahi S, Bettauer V, Khurdia S, Omran RP, Simpson S, Xie JL, Whiteway M, Berman J, Hallett MT | 2022 | Candida albicans exhibits heterogeneous and adaptive cytoprotective responses to anti-fungal compounds | https://www.ncbi.nlm.nih.gov/geo/query/acc.cgi?acc=GSE204903 | NCBI Gene Expression Omnibus, GSE204903 |
| Dumeaux V | 2022 | sc-candida_paper | https://osf.io/5tpk3/ | Open Science Framework, 5tpk3 |

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

## Appendix 1

### Drug diffusion assays with FCZ-treated cells at days 1–6

Diffusion disk assays provide a means to measure tolerance and resistance of a fungal population to a drug (*Gerstein et al., 2016*). In our case, we use these assays to measure tolerance and resistance of *C. albicans* in response to FCZ. Toward this end, two SC5314 colonies were transferred from YPD agar plates to 5 ml YPD liquid media and incubated at 30°C overnight. Cells were washed the next day with PBS 1× twice and an aliquot of $10^6$ cells/ml was transferred into an Eppendorf tube. 100 µl of this suspension was plated on YPD agar and spread evenly with a cell spreader. A FCZ disk (Bio-Rad #62802) was placed at the center of the plate and incubated at 30°C. This was repeated three times (replicates 1–3) for a total of two biological replicates (experiments 1–2).

Images were captured every 24 hr for 6 days (*Appendix 1—figure 1*). *Appendix 1—figure 2* describes how the radius (RAD) and the fraction of growth (FoG) are estimated from these images to provide measures of drug resistance and tolerance, respectively. We use the diskimageR R package to estimate the intensity of 72 lines (in 5° increments around the circle) from the disk edge to the rim of the plate for each replicate across 6 days post-exposure (*Appendix 1—table 1*). A Kruskal-Wallis $\chi^2$ test was used to test whether there was a difference in the mean of either the FoG or RAD between day 1 and the remaining time points versus a null hypothesis of no difference.

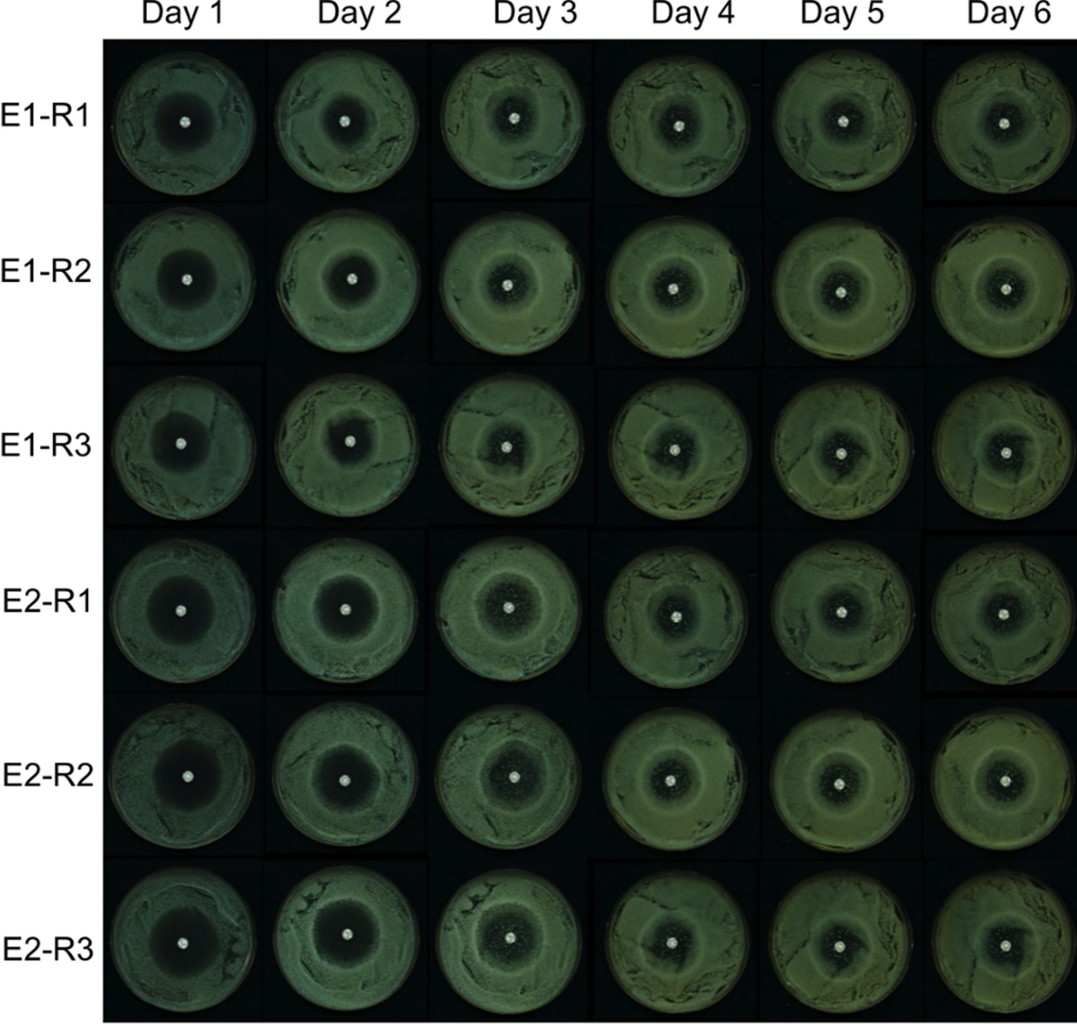

**Appendix 1—figure 1.** Fluconazole (FCZ) disk assay results across 6 days. Two cultures (biological replicates E1 and E2) of *C. albicans* SC5314 were grown in YPD and spread on three plates (technical replicates R1, **R2, and R3**). The bright white spot at the center of the plate is the FCZ diffusion disk.

**Appendix 1—figure 2.** Diffusion disk assay experiments with fluconazole. (**A**) A *C. albicans* culture with fluconazole (FCZ) disk in the middle of plate. The red radial line represents 1 of 72 measurements taken every 5°. (**B**) Top is a blown-up and restricted image of the red line from (**A**) and bottom represents the pixel intensities from 0 (edge of disk) to 40 (edge of plate). (**C**) Gray curves are from the 72 measured radial lines. Black represents the average of these 72 measurements per mm from disk edge (x-axis). Light blue dot is the $RAD_{20}$, defined as the point where there is a 20% reduction in growth. Middle blue dot is the $RAD_{50}$ and dark blue represents the $RAD_{80}$. The $FoG_{20}$ is defined as the fraction of the area under the black line from 0 to the $RAD_{20}$ (i.e., the amount of growth observed) divided by the total potential growth (delimited by dotted line). The $FoG_{50}$ and $FoG_{80}$ are defined analogously (adapted from *Gerstein et al., 2016*).

**Appendix 1—table 1.** Two biological (independent) *C. albicans* SC534 cultures and three technical replicates were subjected to fluconazole (FCZ) diffusion disk assay (DDA) experiments and averaged. The table represents the output from diskimageR over 6 days of imaging. The increase in tolerance, which is measured by the fraction of growth (FoG), from day 1 to day 2 is significant as is the increase from day 2 to day 3 (both p<0.001, Kruskal-Wallis $\chi^2$). There is a slight but statistically reduction in the $RAD_{20}$, suggesting that there has been a small increase in resistance (p<0.05, Kruskal-Wallis $\chi^2$). However, we can see that the vast majority of the culture is not resistant.

| Day | FoG | $RAD_{20}$ |
|-----|------|-------|
| 1 | 0.14 | 16.33 |
| 2 | 0.21 | 14.17 |
| 3 | 0.37 | 12.67 |
| 4 | 0.37 | 13.00 |
| 5 | 0.38 | 12.83 |
| 6 | 0.4 | 13.00 |

## Appendix 2

### Supplemental results

UT cells are found primarily in cluster 3-green and to a lesser extent in cluster 2-pink (*Figure 3A–B*; *Appendix 2—figure 1A*). CSP cells were mostly found in cluster 1-darkpink while RAPA cells were mostly found in cluster 2-lightpink (*Figure 3A and B*; *Appendix 2—figure 1*). The remaining 3% of cells (N=573) were outliers scattered across 19 distinct clusters (others cluster-darkblue; *Figure 3A and B*) that we termed 'comets'. In this section, we will describe the biological characteristics for each of these clusters.

### UT cells show expectedly higher glycolysis gene expression compared to antifungal-treated cells on days 2–3

UT cells have the highest expression of the glycolysis and carbohydrate reserve metabolism pathways (*Figure 3C*, green column; *Appendix 2—figure 1*). This is consistent with the fact that UT cells were collected during log-phase before glucose became a limiting factor, while all non-UT cells were harvested at least 2 days post-media renewal.

### Exposure to FCZ: emergence of 'comets'

In addition to the five clusters that comprise the vast majority of all profile cells, we observe 19 small clusters that are scattered across the UMAP embedding (*Figure 3A*). We term these 19 small clusters 'comets'. Using the sc profiles, we investigated whether there is evidence that the comets are interesting biological phenomena versus stochastic noise from the sc profiling technique. Since the small size of the comets (comets contain 7–126 cells) creates significant technical/statistical challenges, our investigation and findings are built using several distinct analysis methods.

*DEG*s: all comets combined versus the five major clusters. All cells that belong to comets were combined into one 'super comet' and contrasted against the five major clusters to identify DEGs using two different approaches (Materials and methods, 'DGE analysis'; *Appendix 2—figure 2*; *Appendix 2—figure 2—source data 1*). These analyses identified *GCN4* (a transcriptional activator of the general amino acid response), and *ILV2* and *ILV5* (which encode for enzymes involved in branched-chain amino acid biosynthetic pathway) (*Appendix 2—figure 2* ; Bayes factor >2.5 and proportion of non-zero value >0.2). The ILV pathway is known as a fungal amino acid starvation therapeutic target and *ILV5* is recognized as a component in the *C. albicans* amino acid starvation response (*Kingsbury and McCusker, 2010*; *Tripathi et al., 2002*). It is regulated by the transcriptional activator *GCN4* in both *S. cerevisiae* and *C. albicans* (*Hinnebusch, 2005*; *Natarajan et al., 2001*; *Tripathi et al., 2002*).

*Pathway and signature analysis* can sometimes detect more subtle differences in comparison to univariate DEG analysis, especially in cases such as ours where there are few cells available for the analysis. *Figure 3C* (column labeled 'others') summarizes our observations from this analysis, which identifies an enrichment of *GCN4* target genes encoding amino acid biosynthetic enzymes, suggesting that comet cells are undergoing amino acid starvation. There are at least three reasons why *C. albicans* cells may be depleted of amino acids: (i) exposure of *C. albicans* to FCZ reduces the pool size of amino acid biosynthesis intermediates (*Katragkou et al., 2016*), (ii) the cells are exporting amino acids, which is consistent with the observed upregulation of efflux signatures in *Figure 3C* or (iii) the amino acid starvation response could be related to changes in the translational machinery when cells experience extreme stress. This is consistent with the observation of low expression of rRNAs in *Figure 3C*.

GO analysis of the DEGs also revealed an enrichment of genes involved in DNA replication regulation, actin cytoskeleton, plasma membrane organization, and mitotic ring assembly, all highly expressed in comet clusters (*NHP6A, HHF22, HHF1, SBA1, NCE102, KIN2, RVS167, CDC12*) (*Appendix 2—figure 2—source data 1; 2*; *Appendix 2—figure 2*). This is in line with the observation that FCZ affects bud formation and is associated with DNA replication or spindle pole body duplication (*Harrison et al., 2014*). In our prior research, we discovered that within 4–8 hr of drug exposure (at concentrations greater than the MIC), approximately 20% of mother-daughter cell pairs produced a single granddaughter cell, resulting in three cells (*Harrison et al., 2014*). Despite this, the mother and daughter nuclei continued to divide, generating four daughter nuclei. This process led to one cell containing twice the normal DNA content, and the resulting 4N cells frequently underwent abnormal divisions, giving rise to aneuploid offspring (*Harrison et al., 2014*). Using the

sc data, we searched for aneuploidy evidence by calculating the percentage of reads assigned to genes in each chromosome. We found that cells in cluster 16 have a significantly higher proportion of reads assigned to genes in chromosome 2, which could indicate trisomy or local chromosomal amplification (*Appendix 2—figure 2*).

*DEGs*: each comet versus each main cluster. Although the number of cells per comet cluster is small (from 7 to 126 cells), individual molecular marker genes could be identified in some comets. Some of these markers have been reported previously to be involved in FCZ tolerance or resistance (Appendix 2—figure supplement 2B; Materials and methods, 'DGE analysis'; top 5 with Bayes factor >3 and proportion of non-zero value >0.2). These included cell wall organization and adhesion (*ADA2, TPK2, LMO1, KRE9, RDH54, DFI1*) and chromosomal and cytoplasmic division (*AXL2, HOS3, RDH54, RFA2 RIM15, NPL4, CDC5 BUB2*).

In summary, although the comets each contain a small number of cells make it difficult to assess any statements with statistical rigor, comets generally overexpress GCN4 and related genes encoding amino acid biosynthetic enzymes, suggesting the cells are starved for amino acids. Pathway analysis is consistent with various known reasons for amino acid starvation. Specifically, efflux pumps are concomitantly overexpressed and translational machinery is underexpressed. Comets also consistently overexpress DNA replication regulation, actin cytoskeleton, plasma membrane organization, and mitotic ring assembly processes. We have previously established that FCZ exposure is associated with DNA replication or spindle pole body duplication, leading to aneuploidy. At least one comet has strong evidence of an aneuploidy along chromosome 2.

## Subinhibitory dose of CSP triggers the *Rd* response

CSP, a first-line clinical treatment echinocandin, acts by inhibiting β-1,3 glucan synthesis and disrupting the fungal cell wall (*McCormack and Perry, 2005*). Although it is effective against most *Candida* species, a known mechanism of echinocandin resistance involves point mutations in 'hot spot' regions of FS-encoded subunits of a glucan synthase (*Perlin, 2015*; *Pristov and Ghannoum, 2019*). Tolerance to CSP increases with larger amounts of cell wall chitin, potentially due to aneuploidy and chromosome 5 rearrangement (*Yang et al., 2017*). Even though fungicidal CSP and fungistatic FCZ have different modes of action, they share similar tolerance mechanisms associated with Hsp90, the calcineurin pathway (*Singh et al., 2009*) and the pH-responsive RIM pathway (*Garnaud et al., 2018*). To treat a *C. albicans* population, we used a subinhibitory concentration of CSP, well below its MIC$_{50}$ (1 ng/ml), as higher doses led to significant cell aggregation, rendering the cells unsuitable for sc profiling.

On day 2, the majority of cells were found in cluster 1-darkpink, where *Rd* FCZ cells also reside (86% of all CSP, *Figure 3A and B*; *Appendix 2—figure 1*). We examined the differences between CSP and FCZ cells in the *Rd* state and identified only one significantly differentially expressed gene (FDR <0.1), suggesting a highly similar *Rd* response for both antifungal treatments, at least at the selected dosage levels and day 2 time point. Unlike FCZ, no bifurcation was observed for CSP at day 2 (i.e., we did not observe any *Sd* cells), which could be partially due to our choice of a low dosage level, a factor previously shown to be crucial in determining the cellular response to CSP in *C. albicans* (*Enjalbert et al., 2006*; *Enjalbert et al., 2003*). There was only minor heterogeneity in the cellular response to CSP (*Figure 3B*).

Higher concentrations of CSP and longer time courses might induce a bifurcation similar to FCZ, leading to the emergence of *Sd* cells characterized by high expression of heat-shock protein coding genes. Since HSP70 was identified as a strong marker of the *Sd* response, we analyzed microscopy images of our GFP-tagged HSP70 cells exposed to CSP or FCZ for 2 and 3 days. Although a significant fraction of FCZ-treated cells highly expressed HSP70 at days 2 and 3, confirming the presence of *Sd* cells, no similar trend was observed in CSP-treated cells at either time point (*Appendix 2—figure 1*).

In conclusion, cells treated with CSP closely co-clustered with FCZ *Rd* response cells, but there was no evidence of CSP cells in the *Sd* response at day 3. The similarity in the early response is intriguing, considering that CSP and FCZ have distinct mechanisms of action (i.e., disrupting cell wall biosynthesis versus disrupting ergosterol biosynthesis). It will be interesting to determine the universality of the *Rd* response in future antifungal profiling efforts and the potential transition to the proliferative tolerant *Sd* state.

## RAPA survivors exhibit the highest expression of hyperosmotic stress and lowest expression of RPs

The TOR (target of RAPA) pathway controls many processes including protein translation, autophagy, apoptosis, and cell growth in response to nutrient availability (*Bastidas et al., 2009*; *Cruz et al., 2001*; *Heitman et al., 1991*; *Schmelzle and Hall, 2000*). Although *C. albicans* is sensitive to RAPA (*Baker et al., 1978*), it has weak antifungal properties when used alone (*Baker et al., 1978*; *Tong et al., 2021*). However, RAPA treatment in combination with azoles, including FCZ, exhibits a synergistic effect (*Tong et al., 2021*), eliminating drug tolerance (*Rosenberg et al., 2018*). We sc-profiled *C. albicans* populations exposed to a concentration of RAPA that does not visibly impact growth (0.5 ng/ml) but is known to inhibit FCZ tolerance (*Rosenberg et al., 2018*).

Cells treated with RAPA primarily localize to cluster 2-pink and, to a lesser extent, cluster 4-turquoise in the *Sd* state (*Figure 3A and B Appendix 2—figure 1*). We previously noted that *Sd* FCZ cells exhibit high activation of heat-shock stress response genes. A similar trend is observed in RAPA-treated cells, but they mostly display a pronounced hyperosmotic stress response (*Appendix 2—figure 1* ). In fact, this elevated hyperosmotic stress response is present in all RAPA cells, regardless of their cluster classification (*Appendix 2—figure 1* , light blue curves). Specifically, RAPA cells express high levels of the sodium pump ENA2 and the glycerol-3-phosphate dehydrogenase GPD2 (*Appendix 2—figure 1* ). Both genes are orthologs of hyperosmotic stress-responsive genes in *S. cerevisiae* and are induced in *C. albicans* in response to hyperosmotic stress (*Enjalbert et al., 2003*), indicating that hyperosmotic stress gene activation is unique to RAPA treatment.

Although RAPA-enriched regions initially seem to have moderate to low RP levels (*Figure 3F and J*), closer examination reveals that RAPA cells have the lowest RP expression compared to cells from any other condition (*Appendix 2—figure 1*). This is expected, as RAPA's inhibition of TOR signaling is known to decrease RP gene expression (*Powers and Walter, 1999*). Consequently, no RAPA cells are found in the *Rd* state, which is characterized by high RP. When RAPA and FCZ are used together, drug tolerance is eliminated, but drug resistance remains unaffected. Our findings suggest that this might happen due to the inhibition of the *Rd* response, preventing cells from transitioning to the tolerant *Sd* state. Interestingly, RAPA specifically induces the expression of hyperosmotic stress-related genes. Past research has shown that the Tor1 pathway regulates phase separation and the biophysical properties of the cytoplasm by modulating macromolecular crowding (*Delarue et al., 2018*). The hyperosmotic stress response to RAPA exposure could be a result of disrupted cytoplasmic biophysical properties, leading to altered osmotic balance in cells.

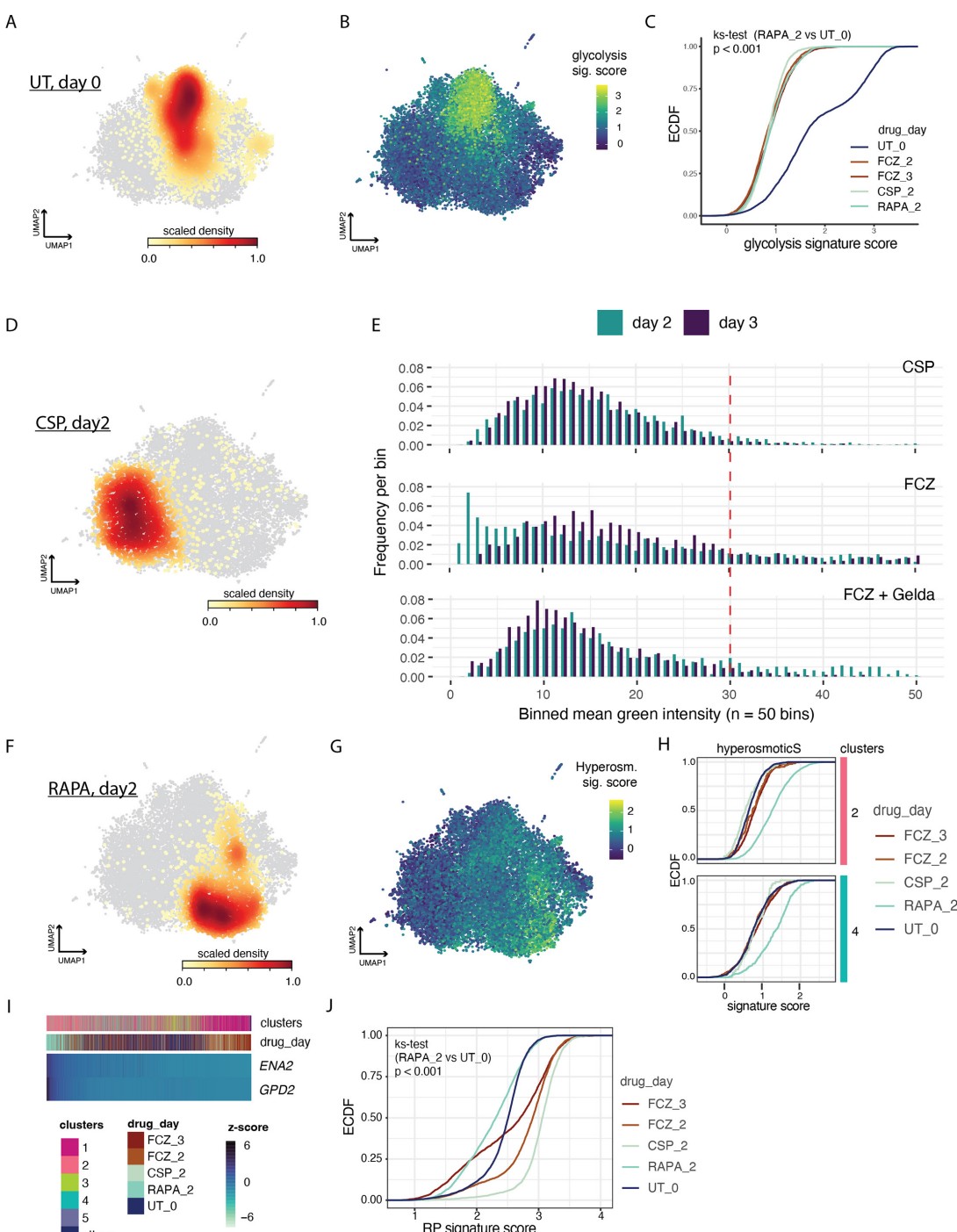

**Appendix 2—figure 1.** Single cell analyses of additional anti-fungal drugs. (**A**, **B**, **D–F**) Uniform Manifold Approximation and Projection (UMAP) embedding from *Figure 3A*. Color bar depicts: (**A**) density of untreated (UT) cells, (**B**) expression score for the glycolysis signature. (**C**) Empirical cumulative distribution (ECDF) of the glycolysis signature scores for each cell in the different conditions (drug_day, color); Kolmogorov-Smirnoff test, p < 0.001. (**D**) Density of caspofungin (CSP) cells at day 2. (**E**) Histograms depicting expression of heat-shock protein 70 (HSP70) in (i) CSP or (ii) fluconazole (FCZ)-treated cells after 2 days (green) 29 and 3 days (blue) exposure. (**F**) Density of rapamycin (RAPA) cells after 2 days of exposure. Cells with high expression of HSP70 fall to the right of the dashed red line. There is a statistically significant depletion of cells highly expressing HSP70 (a marker of Sd; KM test, p < 0.01). (**G**) Score for the hyperosmotic stress signature mapped onto the UMAP embedding from *Figure 3A*. (**H**) ECDF of the hyperosmotic stress (hyperosmoticS) signature score for each cell in clusters 2 (top) and 4 (bottom) for the different conditions (drug_day, color). (**I**) The heatmap depicts genes (rows) in the hyperosmotic stress

*Appendix 2—figure 1 continued on next page*

*Appendix 2—figure 1 continued*
signature with significant variability and consistent expression across cells (columns). We used MAGIC to impute expression levels (z-score, color bar). Cells are linearly ordered by the overall magnitude of gene expression (mean gene rank). 'Clusters' (top) indicates the Leiden cluster of origin for each cell and 'drug_day' indicates the condition for each cell. (**J**) ECDF of signature scores for each cell under the different conditions (color) (ks-test, Kolmogorov-Smirnoff test).

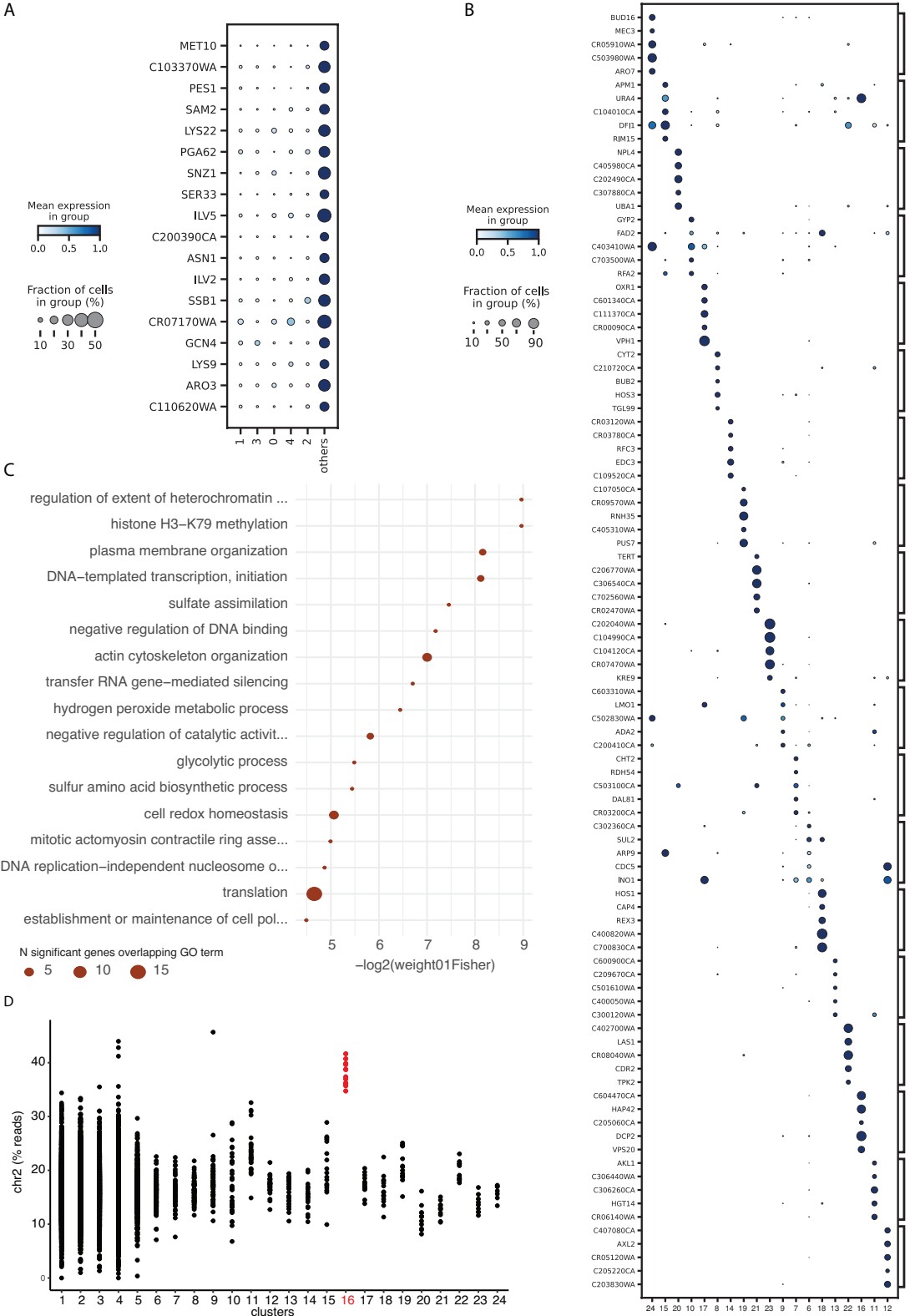

**Appendix 2—figure 2.** Differentially expressed genes and pathways associated with comets. (**A**) Genes overexpressed in the collection of 19 comets compared to other cells in main clusters (Bayes factor > 2.5 and proportion of non-zero value > 0.2). Color bar indicates the mean expression across the cells in each cluster. The size of the dot indicates the proportion of cells which express that gene within each cluster. (**B**) Similar plot than (**A**) for genes differentially expressed in each comet cluster compared to others (Bayes factor > 3 and proportion of

*Appendix 2—figure 2 continued*

non-zero value > 0.2). (**C**) Gene ontology (GO) enrichment for genes identified as differentially expressed in comets compared to other clusters using pseudo-bulk differential expression analysis (60 genes; FDR < 0.1 ; Appendix 2—figure 2—source data 1, 2).The size of the dot is proportional to the number of genes in the list which overlap with the corresponding GO term. (**D**) Percent of reads assigned to genes in chromosome 2 for each cell classified in the 24 distinct clusters.

The online version of this article includes the following source data for appendix 2—figure 2:

**Appendix 2—figure 2—source data 1.** Differentially expressed genes in cells classified in the comet clusters (pseudo-bulk samples) compared to other fluconazole (FCZ)-treated cells (pseudo-bulk samples).

**Appendix 2—figure 2—source data 2.** Gene ontology terms enriched in genes differentially expressed between comets (pseudo-bulk) and other fluconazole (FCZ)-treated cells (pseudo-bulk) listed in Figure 3—table supplement 1A.

