## [Editor Report]

The valuable study by Dumeaux et al. examines the transcriptional response to antifungal treatment in the major opportunistic human fungal pathogen *Candida albicans*. Using solid methodology, including a novel droplet-based single cell transcriptomics platform, the authors report that fungal cells exhibit heterogeneity in their transcriptional response to antifungal drug treatment. The ability to study the trajectories of individual cells in a high-throughput manner provides a novel perspective on studying the emergence of drug tolerance and resistance in fungal pathogens.

---

## [Decision Letter]

**Decision letter after peer review:**

Thank you for submitting your article "*Candida albicans* exhibits heterogeneous and adaptive cytoprotective responses to anti-fungal compounds" for consideration by *eLife*. Your article has been reviewed by 3 peer reviewers, and the evaluation has been overseen by a Reviewing Editor and Naama Barkai as the Senior Editor. The following individual involved in review of your submission has agreed to reveal their identity: David Shore (Reviewer #1).

Essential revisions:

As you will see from the reviewers' individual reports, there was consensus that the manuscript and work reported are interesting but also had concerns about presentation and some of the controls used in the study. Thus, the manuscript will require revision, including potentially additional experiments, and one more round of review.

1) There is concern that there is significant overlap (and lack of clarity) with respect to data previously published (Battauer et al. 2020) and data that are new to this study. It will be imperative to clearly explain where the data in this new manuscript are coming from.

2) There was consensus among reviewers for a revised Introduction section that better sets up the rationale for this study (e.g., Reviewer #2, point #2 and Reviewer #3, point #1).

3) There was concern about the use of different time-points associated with controls vs. treatments (e.g., Reviewer #2, point #4) – addressing this point might necessitate collection of additional data.

*Reviewer #1 (Recommendations for the authors):*

Since the underlying biology here will be of most interest to researchers who are unlikely to be versed in the experimental and analytical methods used throughout, I think that the authors should make a better effort to help the reader begin to understand the bases of their analytical methods and their potential pitfalls. For example, they don't even define the ubiquitously applied "UMAP" method, much less explain in general how it works, what it reveals, and what its limits might be (though it is defined in the McInnes et al. reference title). I suspect that most readers will be unfamiliar with the details of single-cell transcriptomic analysis, which is complex and relatively recent. Μore science-related comments are as follows:

1. The significance of the dozen or so small "comet" clusters is unclear. On line 525 they are referred to as possible "trimeras" (What does that mean? Should it read "trisomes") "…with unstable genomes previously identified after FCZ treatment…which might have arisen as a response to amino-acid starvation." This latter point might be testable by increasing amino acid concentrations in the medium, which might then abolish the formation of some clusters.

2. In the Discussion the authors describe "bulk DNA-level profiling" of untreated cells and four FCZ treated populations (days 2, 3, 6, and 12). They state that attempts to assign the aberrations they observe to either the α or β response were inconclusive, but it's unclear how this would have even been possible to do. Please clarify. It is also stated in the text that variation levels peak at day 3, but it would seem to me to be day 2. Furthermore, on day 2 there would appear to be a nearly uniform increase in signal for all chromosomes shown, suggesting that there might be widespread trisomy. These points all need to be addressed and clarified.

3. The significance of the so-called "trajectory" in Figure 4A (black line) is unclear to me. From the text, it would appear to me that clusters 3 and 4 represent β-like cells on days 6 and 3, respectively. But what are the expression differences between these two groups? A similar question arises for clusters 1 and 2, which are said to be α-like cells. Does the black "trajectory" line imply a strict mode of transition between these 4 states? Why do α-like cells seem to be predominant when longer times are stated elsewhere to favor the β-like state?

4. The authors should state clearly how fast the cultures are growing (bulk measurement) at the time points they examine, and whether they have any way of knowing the growth rate as a function of transcriptome state and time of drug exposure. Related to this, do the authors imagine that α-like cells predominate early simply because they derive from those cells that had the highest levels of RP and ribosome biogenesis gene expression prior to treatment and thus might have been faster growing and/or more capable of escaping the early effects of drug treatment?

5. What is the actual evidence that β cells are derived from α cells through induction of RASTR in the latter population of cells? Related to his, could it be shown that α cells display a higher level of protein aggregation than β cells, perhaps by looking at RP fusions to GFP, versus HSP70-GFP?

6. Can the authors compare in more detail the transcription states of their β cells to the profile of RASTR cells (*S. cerevisiae*) described by Albert et al.? For example, are the targets of Hsf1 known in detail in *Candida albicans*, and if so, how well do they match with the up-regulated genes observed in β cells? Does Hsf1 also target genes encoding components of the proteasome (as in *S. cerevisiae*)?

7. Furthermore, can the authors show that Ifh1 is "condensed" in β cells, as seen clearly using Ifh1-GFP fusions in *S. cerevisiae*? In addition, what is known about Ifh1 targets in *C. albicans* (are they mostly RP genes?), and how well does this group overlap with the profiles of down-regulated genes in β cells? Perhaps the datasets are not robust enough to give this sort of information, but the issues should at least be addressed in the text by the authors.

8. The idea that a kind of persistent RASTR response promotes tolerance is very interesting. Perhaps the authors could test this idea further by asking whether inducing RASTR prior to drug treatment (with diazaborin or an RNA Pol I inhibitor) might strongly increase the fraction of tolerant cells.

9. Line 591: what is meant by "reinitiate translational machinery" with respect to the α cells? And how is this related to their persistence over time?

*Reviewer #2 (Recommendations for the authors):*

1. This manuscript emphasizes the lack of single-cell transcriptomics in *C. albicans* (and fungi broadly), although there does appear to be other work published in this area (Dohn et al. 2021 – which also includes antifungal treatment). Relationship to other work in this area should be more clearly addressed, and differences in findings with regards to antifungal treatment should also be addressed.

2. The introduction to this manuscript is framed around antifungal drug tolerance, and throughout this research, antifungal tolerance is highlighted as a central research question. However, it is not clear how the experimental design of this work addresses antifungal tolerance in any way. Cells are treated with antifungal drugs and subjected to transcriptomic analysis, but this does not represent a distinct 'tolerant' population of fungi that are being analyzed. This either needs to be much more clearly explained and justified, or more likely, the framing of this work needs to be substantially re-assessed.

3. The authors make reference to another study of theirs (Battauer et al. 2020) and suggest that the transcriptomic profiles reported in this work combine profiles from their previous work. This needs to be much more clearly explained. Have parts of this work already been previously published? How is this analysis unique and novel? The title of the previous manuscript seems very similar to that of this manuscript suggesting there might be a substantial overlap between these works.

4. The major analysis in this work compares untreated cells in log phase, to antifungal drug treated cells grown for 2-3 days in antifungals. It is not clear why untreated cells were not grown for the same duration of time as drug-treated cells, which would certainly alter the findings and analysis. It is thus unclear if the transcriptomic responses described in this manuscript truly represent the consequence of drug treatment itself, or are also influenced by the growth state of the cells (which are in stationary phase after 2-3 days of drug treatment). A comparison to untreated cells in stationary phase would be a more appropriate comparison.

5. The work describes the identification of 184 transcripts on average in each cell, which seems like an incredibly small number. Is this in line with other similar single-cell transcriptomic analysis? Does this number of transcripts enable robust conclusions to be drawn on the transcriptional profile of individual cells?

6. While the paper is generally well written, it is written in an extremely technical manner with much methodological detail (as well as discussion) incorporated throughout the main Results section. Major Results sections also lack clear and concise conclusions to help readers understand and interpret the major findings. This diminishes the clarity of the work and makes the manuscript at times quite dense and difficult to fully interpret.

7. It is unclear why stress response pathways are being assessed in untreated cells without any stress exposure. Should this be analyzed in the drug-treated cells instead?

8. In the growth curves in Figure 3 Supp 1a, it seems that both untreated and drug-treated cells take 2-3 days to reach stationary phase. This seems very unusual for cells that replicate quite rapidly under many growth conditions. Is this due to a nutrient-limited media or some other explanation?

9. It seems surprising that extremely different antifungals (cidal vs. static, different cell targets) elicit a very similar transcriptional response based on this analysis (line 345-348). Can this be explained?

10. Section 8-9 on expanding the analysis to day 6 was difficult to interpret in terms of the rationale for the experimental design and how the findings can be interpreted. It is also unclear how solid agar media-based tolerance assays can be extrapolated to liquid media growth assays with drugs, as these are very different conditions. It is also unclear if there is a day 6 untreated control that is being assessed.

a. The microscopy in associated Figure 4 K is very difficult to see and lacks scale bars.

*Reviewer #3 (Recommendations for the authors):*

Below I provide a few suggestions for how the authors may improve the manuscript, but overall I found it well done.

1. First, it is my understanding that tolerance is defined as survival, but not growth, after exposure above the MIC for drugs that normally would kill most of the population of cells (for non-static drugs). This is the definition put forward in Balaban et al. ("Definitions and guidelines for research on antibiotic persistence"; Nat Rev Microbiol; 2019). I would have called what the authors describe as phenotypic resistance, i.e., a sub-population of cells that can continue to grow after exposure above the MIC, and a non-genetically heritable state (i.e., readily reversible). This is a known phenomenon for some antimicrobials/species combinations, as the authors describe. I think it would be good to standardize usage of the word tolerance through the manuscript, or the authors can better explain how the phenomenon they observe is consistent with Balaban et al.'s definition of tolerance. This is an important semantics issue. Second, the drug concentrations used were at or well-below their MICs. This raises the question of whether the authors are studying the clinical phenomenon that they describe as tolerance (i.e., above the MIC) when the experiment was conducted below the MIC. While I understand their reasoning for using low drug concentrations, and I think there is still something to be learned at these concentrations, I think the authors should better contextualize the clinical relevance of their findings with the caveat that the experimental results were probably collected at sub-clinical concentrations. Admittedly, drug concentration is dynamic during treatment, so at some point the concentration in vivo likely passes through the author's choice of in vitro concentrations. It would be interesting to know how much their results are invariant to the drug concentration chosen.

2. The author's interpretation of the "comets" is questionable. On L244 they say, "This pattern suggests that the small set of cells from each comet have strong transcriptional similarity but each such comet is transcriptionally distinct from the other comets." I do not see how the cells within comets share "strong" transcriptional similarity because they are still spread out. Rather, the comet tails could be interpreted as lineages of descent, wherein each cell becomes more distinct (along some transcriptional axis) than its ancestor. This pattern is commonly observed in genetic data plotted with PCA, which is somewhat related to what is being shown here. Therefore, an alternative interpretation is that some lineages are moving toward a transcriptionally distinct state and leaving behind progeny along their trajectory, especially given that the experimental setup may not quickly remove ancestral cells in the two day exposure. Another explanation is that the comets are composed of a different (minority) morphology, such as filamentous growth. Another (less?) plausible interpretation is that these are noise clusters due to the inherent stochasticity involved in single-cell analyses. I see no reason why these explanations aren't equally reasonable to the author's explanation. Perhaps the authors can incorporate these alternative into their interpretations, but (at a minimum) there needs to be more justification given for the claim of "strong" intra-cluster transcriptional similarity.

3. L434: I am skeptical of the author's interpretation here about the relative fitness of α and β states. It could also suggest switching rates between phenotypic states are different between α and β populations. Also in this paragraph (L439), the authors use OD600 to show a higher growth rate for β over untreated. However, β also reaches a higher yield, which is unexplained. Having done many growth rate experiments with OD600 data, I am always skeptical of drawing large inferences about fitness from OD600. A better test of fitness is a competition experiment. I suggest the authors remove or downplay their conclusions about relative fitness here. Also, why isn't OD600 plotted in log-space (log(OD600)) if the goal is to see the difference in growth rate?

4. The authors did not seem to analyze feature importance in Leiden clustering, preferring to look at cluster associations (z-scores) instead. Leiden clustering is not guaranteed to be optimal, so some understanding of cluster stability would have been useful. That is, if inputs changed somewhat (e.g., subsampling or resampling), would the identified clusters have been similar? This is particularly important given that the UMAP1 versus UMAP2 clusters look like blobs that are split along arbitrary axes. It seems doubtful these clusters are stable, yet the whole manuscript analyzes them as though the blobs are discretized correctly.

5. I have some concerns about the comparison between untreated and treated at the same (or similar) time points. If the transcript profile changes over time, as presumably it would, then it is reasonable to expect that the treated and untreated cells are at different *equivalent* times because their growth rates are different. I have a similar concern with comparing different time points (days 3 versus 6) because fresh medium (YPD) was used in this experiment. How much are transcript abundances static over time? This should minimally be discussed further in the manuscript. Also, the conclusions in L472-475 and L579-592 may need this caveat mentioned.

---

## [Author Response]

Essential revisions:1) There is concern that there is significant overlap (and lack of clarity) with respect to data previously published (Battauer et al. 2020) and data that are new to this study. It will be imperative to clearly explain where the data in this new manuscript are coming from.

We understand this concern and we apologize for the lack of clarity in our previous version of this manuscript. We have adjusted the text in the introduction (and early Results section) to better explain this relationship. Briefly, we originally released the Bettauer et al. BioRxiv in 2020 (https://doi.org/10.1101/2020.01.21.914549), although we were unsatisfied with the descriptive nature of the single cell results, which primarily focused on the identification of the α and β responses. Our rationale was that two senior graduate students were set to finish their degrees, and we thought that the underlying technical contribution alone towards fungal single cell approaches was in and of itself sufficient to form a preprint and useful for the community (several other fungal groups had expressed interest in using our technology and we did not have the capacity ourselves to assist them directly).

However, we continued to do additional validation work (the fluorescence microscopy, CRISPR-mediated knockouts, DNA-sequencing and additional single cell RNA-sequencing) and this new data gave us deeper insight into the nature of the α and β responses. Bettauer et al. (2020) bioRxiv was not published elsewhere. Instead this data was merged with new data into this current effort.

Lastly, when we prepared for the first submission to *eLife*, we attempted to modify the Bettauer et al. (2020) preprint, but bioRxiv was tardy in re-opening the manuscript (we waited several weeks). We felt we could not delay submission any longer and submitted a new preprint instead: Dumeaux et al. (2022) bioRxiv (https://doi.org/10.1101/2022.07.20.500774). This data and analyses in this preprint are supersets of what appeared in Bettauer et al. (2020). We have asked that bioRxiv link this 2022 preprint to the original 2020 preprint. [Changes located at: L102-109; L116; L146-148].

2) There was consensus among reviewers for a revised Introduction section that better sets up the rationale for this study (e.g., Reviewer #2, point #2 and Reviewer #3, point #1).

We have re-worked the entire Introduction addressing specific comments from the Reviewers. We have provided more background and clearer definitions regarding fungal drug tolerance [Changes L53-61]. We have also sought to better explain the rationale for aspects of our study throughout this Response to Reviewers and in the manuscript (for example, for UT cells L184-187). We also moved a comparison of UT cells to the FCZ day 2 and 3 cells to Appendix 2 [Changes L 822-833]; this short comparison is not central to the main findings of the manuscript and has caused confusion as to our primary goal. We really only performed this comparison as a sort of “sanity” check to see if the single cell (sc)-profiling would detect differences between UT and drug treated cells.

3) There was concern about the use of different time-points associated with controls vs. treatments (e.g., Reviewer #2, point #4) – addressing this point might necessitate collection of additional data.

We fully agree with the concerns raised by the reviewers, and we have sought to better clarify throughout the manuscript our choice for controls. Our presentation within the previous version of the manuscript led to some confusion and we realize that we did not fully explain nor justify our choices in these dimensions. We address these issues below in response to specific questions from the reviewers and new data which was generated for this re-submission. [Changes L142-149 and several other sentences throughout the manuscript.]

A general note regarding this re-submission: we realized that the use of α and β to describe the two central subpopulations might be confusing to the *C. albicans* community, since α and a are used to describe the two mating types in this organism e.g. a/α, α/α, a/a. Globally now we refer to the α subtype as *Rd* (Ribo-dominant) and the β subtype as *Sd* (Stress-dominant) to avoid causing confusion in the literature.

Reviewer #1 (Recommendations for the authors):Since the underlying biology here will be of most interest to researchers who are unlikely to be versed in the experimental and analytical methods used throughout, I think that the authors should make a better effort to help the reader begin to understand the bases of their analytical methods and their potential pitfalls. For example, they don't even define the ubiquitously applied "UMAP" method, much less explain in general how it works, what it reveals, and what its limits might be (though it is defined in the McInnes et al. reference title). I suspect that most readers will be unfamiliar with the details of single-cell transcriptomic analysis, which is complex and relatively recent.

We have revisited all aspects of the manuscript and inserted intuitive descriptions of the different methods used here. This includes the concepts of scVI UMAP embeddings, Leiden clusterings, pathway enrichment scores and trajectory analysis [L250-257, L368-373, L699-707]. We detail the location of additional changes below.

Μore science-related comments are as follows:1. The significance of the dozen or so small "comet" clusters is unclear. On line 525 they are referred to as possible "trimeras" (What does that mean? Should it read "trisomes") "…with unstable genomes previously identified after FCZ treatment…which might have arisen as a response to amino-acid starvation." This latter point might be testable by increasing amino acid concentrations in the medium, which might then abolish the formation of some clusters.

We begin by noting that this section on “comet” clusters has been moved to Appendix 2 Supplemental Results as we feel that these observations are not critical to the manuscript and remain largely interesting observations [Changes L893-956]. The section has been completely re-written and clarification of the trimeras term is provided. Specifically, we were referring to our previous study in Harrison et al. where we observed three connected cells composed of a mother, daughter and granddaughter bud, termed trimeras, were observed within hours of FCZ exposure and were shown to give rise to aneuploidies at high frequency. This concept is not strictly necessary for this manuscript except to say that we do observe evidence of the associated aneuploidies [Changes L837-897].

It is an interesting idea to test if increased concentrations in the medium would ablate the appearance of such comets, and we certainly plan to explore this in the future. It is, however, somewhat tangential to the current effort and due to cost issues we haven’t explored this further to date. Disappearance of the comets might indicate that the GCN4-mediated amino acid starvation pathway is central to the formation of aneuploidies and/or provide insight into the rapid evolution of resistant micro-colonies post FCZ exposure. This in itself would constitute a significant manuscript.

Please also see our response to Reviewer #3, Comment 3 which touches on similar issues.

2. In the Discussion the authors describe "bulk DNA-level profiling" of untreated cells and four FCZ treated populations (days 2, 3, 6, and 12). They state that attempts to assign the aberrations they observe to either the α or β response were inconclusive, but it's unclear how this would have even been possible to do. Please clarify.

Before proceeding, we mention here that we decided in light of your comment to remove this paragraph from the Discussion section. Originally we intended to indicate that we did look at issues of chromosomal aberrations, which are common to *C. albicans* especially under stress, but our bulk DNA profiling was inconclusive. The sum of our data, analysis and validation experiments really require a full independent manuscript and the brief mention we make here is likely to only cause confusion.

To answer your technical question: The genes within each observed amplified genomic region were identified. We then examined whether those genes were simultaneously expressed more in either the α*/Rd* or β/*Sd* subpopulations according to the single-cell transcriptomics. The hope here was to find that a set of amplified genomic regions which were statistically enriched for genes that were over-expressed in exactly one of the α or β subpopulations. Since both the α*/Rd* and β/*Sd* subpopulations could “contribute” amplified genomic regions (and therefore over-expressed transcripts for their corresponding genes), we could perhaps assign some genomic regions uniquely to either the α or β subpopulations. We weren’t however able to conclude anything with statistical rigor here.

It is also stated in the text that variation levels peak at day 3, but it would seem to me to be day 2.

Yes indeed. This was a typo.

Furthermore, on day 2 there would appear to be a nearly uniform increase in signal for all chromosomes shown, suggesting that there might be widespread trisomy. These points all need to be addressed and clarified.

We agree. Note however the trimera occurrences described by Harrison et al. are not the only mechanism by which instability is observed in *C. albicans* in response to (drug) stress. In particular, the seminal papers by Ford et al. (doi: 10.7554/*eLife*.00662), Selmecki et al. (2009) and Todd and Selmecki (2020) (cited in the Introduction) describe other mechanisms for aneuploidy and loss across *C. albicans*. This includes recombination/amplification events involving specific-sequence motifs found across the *C. albicans* chromosomes. What we observe here may be due to other mechanisms besides the trisomy. For example, we did perform some ad hoc analysis comparing the predicted sites from Todd/Selmecki with regions of the *C. albicans* genome that were amplified in this study. Straightforward statistical approaches (based on the hypergeometric tests for enrichment) suggest that there is significant (albeit imperfect) overlap with their predictions. We also compared our regions of amplification with those of Ford et al. (see above) and again found significant, albeit imperfect, overlap. This suggests to us that there may be multiple mechanisms “controlling” genome instability in the *C. albicans* response to stress. The strength of amplification (black ticks) for chromosomes 4, 5, 6 and 7 are dramatic compared to other chromosomes (which appear closer to the pattern observed in the untreated samples), with only a few sporadic regions of amplification or loss. We have added to the text to clarify this analysis.

3. The significance of the so-called "trajectory" in Figure 4A (black line) is unclear to me. From the text, it would appear to me that clusters 3 and 4 represent β-like cells on days 6 and 3, respectively. But what are the expression differences between these two groups? A similar question arises for clusters 1 and 2, which are said to be α-like cells. Does the black "trajectory" line imply a strict mode of transition between these 4 states? Why do α-like cells seem to be predominant when longer times are stated elsewhere to favor the β-like state?

We note that we have re-written these paragraphs entirely to improve their readability. Changes Figure 4 is now Figure 5 pg 16. Changes in text L334-373. Discussion about the trajectory and meaning: [L368-373]

Indeed, clusters 3 and 4 represent β/*Sd* cells on days 6 and 3, respectively. First, we should emphasize that there are quite a few genes that are differentially expressed between β/*Sd* cells on days 6 and 3. The main differences are highlighted on L354-367 and Figure 3G.

In contrast, there are no significant robust difference for α*/Rd* day 3 versus day 6 (cluster stability analysis, Figure 5B).

Trajectory analysis based on single-cell RNA profiles allowed us to order cells along a “differentiation” trajectory. We now inserted intuitive descriptions for trajectory analysis in the manuscript; the essential idea is to consider the transcriptional profile of a cell as a sort of single still image from an animation. If the trajectory analysis (based on a tool called Slingshot) can order the still images to make a complete animation (denoted by the black lines), then this is considered one type of evidence that, when one population is adjacent to a second population (black edge), then the first population is derived from, or at least related to, the second population. In the end, the trajectory helps to confirm that β/*Sd* day 6 cells have evolved from β/*Sd* day 3 cells. For α*/Rd*, the lack of any robust transcriptional changes would already suggest that day 6 cells evolved from day 3 cells.

We think that there may be some confusion w.r.t. the last comment, as *α/Rd* cells are predominant at day 3 but β/*Sd* cells are predominant (or at least equal) at day 6. We speculate as to why β/*Sd* become more predominant on lines L364-367 and in the Discussion L393-399, L413-417.

4. The authors should state clearly how fast the cultures are growing (bulk measurement) at the time points they examine, and whether they have any way of knowing the growth rate as a function of transcriptome state and time of drug exposure. Related to this, do the authors imagine that α-like cells predominate early simply because they derive from those cells that had the highest levels of RP and ribosome biogenesis gene expression prior to treatment and thus might have been faster growing and/or more capable of escaping the early effects of drug treatment?

First, please note that we re-did the growth assays of cultures subjected to the different drugs at the relevant time points (Figures 3 figure supplemental 1B, Figure 5I). The cultures harvested for the untreated analysis were certainly in log phase; the growth curves suggest that the samples for all other time points are in stationary phase.

We do not find any evidence of a subpopulation of untreated cells that had higher RP and ribosome biogenesis expression and therefore we do not think that this is why the α*/Rd* cells predominate early. Our data suggests that the α*/Rd* cells only appear post treatment, suggestive of a “quick” acute response to drug exposure causing the dysfunction of the translational machinery.

5. What is the actual evidence that β cells are derived from α cells through induction of RASTR in the latter population of cells? Related to his, could it be shown that α cells display a higher level of protein aggregation than β cells, perhaps by looking at RP fusions to GFP, versus HSP70-GFP?6. Can the authors compare in more detail the transcription states of their β cells to the profile of RASTR cells (*S. cerevisiae*) described by Albert et al.? For example, are the targets of Hsf1 known in detail in Candida albicans, and if so, how well do they match with the up-regulated genes observed in β cells? Does Hsf1 also target genes encoding components of the proteasome (as in *S. cerevisiae*)?

Our primary sources of evidence are the transcriptional profiles of α*/Rd* and β/*Sd* now presented in more detail in Section 5 of the Results. We expanded our analyses and have re-written some text to better express our core findings [changes L305-329]. One source of evidence for this relationship originates from analyses of the list of RASTR genes (and its associated processes, Table 1) from the literature and how they are expressed in either the α*/Rd* and β/*Sd* subpopulations. To investigate your questions in more detail, we identified *C. albicans* orthologs of *S. cerevisiae* RASTR genes (gene list supplied by B. Albert) together with a list of constitutive Hsf1 target genes identified in *C. albicans*
(Leach et al., 2016). These genes, and their direction of expression, are now listed in Table 1A and B. Note for example that β/*Sd* cells exhibit decreased gene expression of ribosome processing (RP) genes (Figure 3E-F), consistent with the RASTR list in Table 1A (from B. Albert).

We also investigated whether β/*Sd* cells have enhanced expression of genes that are upregulated in RASTR by developing a signature score presented in Figure 4A (Albert list). Figure 4A shows that the right side of the UMAP, which is the location of the β/*Sd* cells, more highly express the genes listed as upregulated by Albert at al. (protein folding, proteolysis, glucose and pyruvate metabolic process, and others). Similarly, Figure 4B shows the expression of Hsf1 target genes from Leach et al. Again, the β/*Sd* cells show heightened expression of the constitutive targets of Hsf1. This comparison alone is not sufficient to argue that β/*Sd* cells are derived from the α*/Rd* cells. We can only hypothesize that β/*Sd* cells, which express all the hallmarks of RASTR activation, would survive and assume a transcriptional profile consistent with RASTR. At this time point, we do not know how we could track individual cells (e.g. via barcodes) and perform single-cell profiling to establish this dynamic behavior. The reviewer is correct to question our implication of causality. We have modified the text in the last section of the Results and Discussion to clarify this.

The observation that α*/Rd* cells predominate early (day 2) but there is an increase in the relative fraction of β/*Sd* cells at day 3 and day 6 also at least suggests a model whereby RASTR is “saving” α*/Rd* cells and sending them to β/*Sd* heaven.

At this stage, we do not know how to directly test for the relative degree of protein aggregation of α*/Rd* versus β/*Sd* cells. We had difficulties to establish a good dual fluorescent strain to measure both aggregation in the Rd and Sd cells simultaneously. We do observe evidence of aggregation, but we are unable to strictly limit this to α*/Rd* and not β/*Sd* cells. We do want to stress here that α*/Rd* transcriptional profiles are very distinct with overexpression of many genes almost exclusively related to ribosome biogenesis, rRNA processing and protein translation [L285-290; Figure 3 —figure supplement 2B; Figure 3 – source data 1,2].

7. Furthermore, can the authors show that Ifh1 is "condensed" in β cells, as seen clearly using Ifh1-GFP fusions in *S. cerevisiae*? In addition, what is known about Ifh1 targets in C. albicans (are they mostly RP genes?), and how well does this group overlap with the profiles of down-regulated genes in β cells? Perhaps the datasets are not robust enough to give this sort of information, but the issues should at least be addressed in the text by the authors.

These are excellent suggestions. Unfortunately, the Ifh1 gene is not well represented in our sc-transcriptomics data, and the meager detected expression is not significantly different between the α*/Rd* and β/*Sd* subpopulation. We focused instead on the expression profile of targets of Ifh1.

– Wade et al. (PMID: 15616568) state only that the vast majority of targets are RP genes. As per Figure 3F**,** we know that the RP genes are high in the α*/Rd* subpopulation and very low in the β/*Sd* subpopulation, consistent with the RASTR sensing mechanism.

– We examined Ifh1 targets provided by PathoYeastract (C.albicans http://yeastract-plus.org/pathoyeastract/calbicans/index.php) (n = 148 genes incl. 41 putative genes and 64 RP-encoding genes). Among those, 103 are detected in our profiles. We selected the most variable genes across cells (n=74) and plotted their expression in the heatmap in Author response image 1. Most targets are more highly expressed in α*/Rd* cells (darkpink cluster1) than in β/*Sd* cells (turquoise cluster4): this includes many RP-coding genes as well as *YST1* (ribosome associated protein gene), *ARF2* (stress response) and *UBI3* (alias RPS31). The rest of the IFH1P targets (n = 10 at the bottom of heatmap) are most highly expressed in cluster (green cluster 3) enriched for untreated cells; they are not differentially expressed between the α/*Rd* and β/*Sd* subpopulations except for *CCP1* (stress response), *ERG251* (ergosterol biosynthesis), and *ADE17* (adenine biosynthesis). These genes are present in the list of genes differentially expressed between α*/Rd* and β/*Sd* provided in Figure 3 – source data 1.

Overall, this analysis further confirms that RP-encoding genes are highly expressed in α*/Rd* cells (and lowly expressed in β/*Sd* cells). Since this is well established in the manuscript already, we choose to restrict this figure to Author response image 1 only. We have added text to discuss Ifh1. [Changes L326-329]

**Author response image 1. sa2fig1:** The heatmap depicts the expression of the most variable IFH1 targets (rows) across cells (columns). We used MAGIC to impute expression levels (z-score, color bar). Cells are linearly ordered by the overall magnitude of gene expression (mean gene rank). “Clusters” (top) indicate in which cluster the cell is assigned. Recall that cluster 1 (red) corresponds to α*/Rd* and cluster 4 (turquoise) corresponds to β/*Sd*. Cluster 3 corresponds to untreated cells.

8. The idea that a kind of persistent RASTR response promotes tolerance is very interesting. Perhaps the authors could test this idea further by asking whether inducing RASTR prior to drug treatment (with diazaborin or an RNA Pol I inhibitor) might strongly increase the fraction of tolerant cells.

This is an excellent idea and we very much intend to do this and a series of additional supporting experiments with appropriate controls. After extensive discussion with our team, we judged that a proper in depth investigation of this would require a very significant and expensive effort. Although the results would add more mechanistic understanding/validation to this manuscript, it is currently not financially feasible for my lab and would delay publication of this already lengthy manuscript by another year. We truly hope that the reviewer understands our constraints and the costs associated e.g. with the sc-profiling.

9. Line 591: what is meant by "reinitiate translational machinery" with respect to the α cells? And how is this related to their persistence over time?

Thank you for pointing this out – we have removed this idea from the manuscript.

Reviewer #2 (Recommendations for the authors):1. This manuscript emphasizes the lack of single-cell transcriptomics in C. albicans (and fungi broadly), although there does appear to be other work published in this area (Dohn et al. 2021 – which also includes antifungal treatment). Relationship to other work in this area should be more clearly addressed, and differences in findings with regards to antifungal treatment should also be addressed.

We have adjusted the Introduction to better cover the results of Dohn et al. However, we would like to note that the original bioRxiv (2020) version of this paper appeared well before Dohn et al. in 2021. It is an unfortunate oversight that Dohn et al. does not cite our effort as microbial DROP-seq (mDROP-seq) has similarities to our published protocol. Nevertheless, there are many differences in the tools and parameters between our two efforts with respect to the quality control and pre-processing of sc-transcriptomics data, yet the number of genes/transcripts per cell were quite comparable.

The results in Dohn et al. support our findings, including those reported in our original 2020 bioRxiv preprint. To summarize their results:

– A “mixed sample” of *S. cerevisiae* and *C. albicans* was DROP-seq’d. They could differentiate between these two fungi. This is reminiscent of the first DROP-seq experiments from Macosko et al. (2015) Cell where mixed mouse and human samples were used to establish the technical efficacy of their system. A nice experiment but independent of any of our results here.

– They examine *S. cerevisiae* subjected to heat shock. This is not related to our manuscript and is similar in structure to several previous single cell profiling efforts that looked at heat shock in *S. cerevisiae*, including the series of papers from the Gasch and later McClean labs. A nice experiment but we do not examine *S. cerevisiae* here so we cannot compare.

– They contrast control *C. albicans* at 1.5 hours and 3 hours post exposure to FCZ. We examine *C. albicans* populations at 2, 3, and 6 days post treatment. Although these are very different time scales, there is still some overlap between our differentially expressed genes (e.g. *WH11* which is discussed in Bettauer et al., 2020). Their analysis of the cell cycle relies on older bulk transcriptional profiles from Spellman et al. By contrast, we combined several sources of evidence including cell cycle related information obtained from more modern transcriptomics data. It is hard to “align” our data with their data for these reasons. At 1.5-3 hours post treatment, Dohn et al. appear to have captured an early cellular stress response. The trajectory analyses mostly point to differences in cell cycle. [Changes located at L102-108]

2. The introduction to this manuscript is framed around antifungal drug tolerance, and throughout this research, antifungal tolerance is highlighted as a central research question. However it is not clear how the experimental design of this work addresses antifungal tolerance in any way. Cells are treated with antifungal drugs and subjected to transcriptomic analysis, but this does not represent a distinct 'tolerant' population of fungi that are being analyzed. This either needs to be much more clearly explained and justified, or more likely, the framing of this work needs to be substantially re-assessed.

We thank the reviewer for pointing out aspects of the introduction that were not clear. We indeed designed the study to focus on antifungal tolerance, by using drug concentrations at and just above the MIC (MIC of *C. albicans* lab strain SC5314 is 0.25-0.5 µg/ml fluconazole) and by analyzing cells at 48 h of drug exposure (a time at which the differences between tolerant and non-tolerant cell growth is evident) (Rosenberg et al. 2018; Gerstein et al. 2016). We now address this by highlighting several aspects of tolerance: First, when tolerance appears, some cells are dividing much more than others. The dividing cells are those exhibiting tolerance to fluconazole, which is a fungistatic drug. We now highlight this, as it is a major distinction between the definition of fluconazole tolerance in fungal cells being treated with a fungistatic drug and the definition of antibacterial tolerance in bacterial cells being treated with a bactericidal drug. The question we are asking is whether there are major distinct subpopulations of cells exhibiting different responses to drug within a single population of isogenic cells. The answer to this is clearly yes, especially with respect to fluconazole, and the two major types of cells are those with an α*/Rd* and those with a α*/Rd* and β/*Sd* response pattern. [Changes L53-73].

3. The authors make reference to another study of theirs (Battauer et al. 2020) and suggest that the transcriptomic profiles reported in this work combine profiles from their previous work. This needs to be much more clearly explained. Have parts of this work already been previously published? How is this analysis unique and novel? The title of the previous manuscript seems very similar to that of this manuscript suggesting there might be a substantial overlap between these works.

Please see our response to these questions above (Essential Revisions #1 and the changes located at L96-99; L109;-114; L137-141). Briefly here, the first preprint from the Hallett lab was never published beyond bioRxiv. Here, this manuscript uses that original bioRxiv data in addition to new data including: more replicates of the sc-transcriptional profiles, as well as microscopy, CRISPR knockouts, DNA-sequencing and disk assays. This manuscript goes significantly further than the first bioRxiv paper, which primarily detailed the technical aspects of the sc-RNAseq assay and identified the existence of the α*/Rd* and β/*Sd* survivor subpopulations after FCZ treatment. Several additional authors were added to this manuscript to assist us with the interpretation of the data (for example, the Berman team, which provided antifungal drug tolerance expertise and performed some of these additional studies).

4. The major analysis in this work compares untreated cells in log phase, to antifungal drug treated cells grown for 2-3 days in antifungals. It is not clear why untreated cells were not grown for the same duration of time as drug-treated cells, which would certainly alter the findings and analysis. It is thus unclear if the transcriptomic responses described in this manuscript truly represent the consequence of drug treatment itself, or are also influenced by the growth state of the cells (which are in stationary phase after 2-3 days of drug treatment). A comparison to untreated cells in stationary phase would be a more appropriate comparison.

There appears to be a misunderstanding. The untreated cells were in log phase, which allowed us to define cell-cycle-specific gene expression patterns. The tolerant cells exposed to drug were still growing, but much more slowly, and they continued to grow between days 2 and 3 of the experiment (Figure 3 —figure supplemental 1B). Between days 3-6, the cells were returned to medium without drug so that recovery could be documented, although in this case, because they had three days to recover, the cells were well past log phase when they were analyzed. It is true that many factors beyond the drug may affect heterogeneity. We think that the interesting observation in these results is that, even after 3 days of recovery, the α*/Rd* and β/*Sd* split persisted. In addition, we note that the α*/Rd* and β/*Sd* cell clusters are defined solely within each culture and neither by contrasting cultures between time points nor by comparing UT versus drug treatments.

We have made more clear in the Methods and in the manuscript in several places e.g. [L196, L509], that state of the cells at time of profiling for each drug/day. We also re-perform the growth curves (Figure 3 figure supplemental 1B and Figure 5I) to provide the reader with a more quantitative estimate of the state of these cells at time of profiling.

5. The work describes the identification of 184 transcripts on average in each cell, which seems like an incredibly small number. Is this in line with other similar single-cell transcriptomic analysis? Does this number of transcripts enable robust conclusions to be drawn on the transcriptional profile of individual cells?

Indeed, expressed in this manner, 184 transcripts per cell on average appears to be a very small number. However, we strongly caution the reviewer from concluding too much based only on the first moment (mean) of a distribution in this context. First, note that there is a very long right tail representing cells with far more transcripts; these cells strongly influence the downstream analyses.

Specifically, every sc-transcriptomics method begins with a quality control pipeline that selects “high quality” cells. The definition of high quality is based on approximately 10 underlying parameters that are determined by different types of *ad hoc* statistical analyses and specific bioinformatics tools (e.g. EmptyDrops software that seeks to determine thresholds of detected RNA that represent successful co-capture of a cell in a droplet versus sequencing of ambient RNA ubiquitously present in the suspension). There are many ways to increase the mean transcripts per cell at the expense of throwing out some sparse cells; however these sparse cells may contribute useful information to the overall patterns in the data. For example, key markers of the α*/Rd* and β/*Sd* classes may be expressed in these “sparse cells” and assist to delineate between subpopulations.

This leads to the question: If you cannot maximize both the number of cells you sequence and the number of transcripts you sequence per cell, which would you choose? It turns out that it is better to increase the number of cells (Zhang, Ntranos, Tse (2020) Nature Communications), even though this may seem counterintuitive at first. A standard approach in sc studies is to first use the sparse sc-data to identify subpopulations. Once identified, we can then pool the data within each subpopulation (ignore barcodes) and do traditional differential analysis. This is exactly what we did in this manuscript: we used the sparse single-cell profiles to discover the α*/Rd* and β/*Sd* subpopulations. We then used traditional bulk analysis by simply binning cells as either α*/Rd* or β/*Sd* (or neither) and then ignored the cell barcodes. This approach provided pseudo-bulk RNA-seq profiles of the α*/Rd* and β/*Sd* subpopulations that were quite robust. It also allowed us to drill down into the genes that are differentially expressed between the two subpopulations.

As we are the first to do sc sequencing in *C. albicans* at large scale, it is a challenge to establish comparisons with previous studies. A study in Saccharomyces performed 11 replicates of their samples with a commercial 10x Inc system (Jackson et al. (2020) *eLife*). Nevertheless, they had a very similar number of transcripts per cell to our data – we captured a maximum of 5956 transcripts/cell and they captured a maximum 5437 transcripts/cell. Note, however, that the lower bound of transcripts/cell is a function of many parameters and, if we repeated the DROP-seq nine more times, we could discard cells to raise our lower bound to an equivalent of 735 transcripts per cell reported by Jackson et al. At the time the experiments were done, each run on the Chromium 10x system was approximately 10x more expensive than a run using our system.

The numbers from Dohn et al. were almost identical to the numbers in this manuscript, although there are differences in many parameters: concentration of beads used, number of DROP-seq runs combined before sequencing, depth of sequencing, quality control parameters etc. Overall, they profile a smaller number of cells (however we consider multiple drugs and more time points). However, it does really come down to Zhang et al.’s observation above – it is more important to increase the number of cells analyzed, than to capture more transcripts per cell. Our goal was to show proof or principle for fungal sc-transcriptional profiling at a reasonable cost, and we think we have achieved that: we were able to identify and partially validate subpopulations that can be explored more deeply.

We also note that section 2 of the Results describes comparisons with bulk transcriptional profiling (Methods 5) and bulk differential expression analyses (Methods 9), which support the idea that the sc-profiling is largely detecting the same transcripts (~6700 in common, only 172 that differ).

In conclusion, whether 184 transcripts per cell (for our specific choice of thresholds and parameters) is or is not sufficient depends on the complexity of the samples, the types of questions that are investigated and the ability to independently validate the existence of subpopulations. Here, for FCZ treated cells for example, we only needed sc-profiles to be of sufficient quality to separate cells into the α*/Rd* and β/*Sd* subpopulations. Improvements to the technology would allow us to identify rarer subpopulations.

6. While the paper is generally well written, it is written in an extremely technical manner with much methodological detail (as well as discussion) incorporated throughout the main Results section. Major Results sections also lack clear and concise conclusions to help readers understand and interpret the major findings. This diminishes the clarity of the work and makes the manuscript at times quite dense and difficult to fully interpret.

We understand and have made an effort to ensure that each section contains intuitive, non-expert explanations of the main findings. We also moved some of the more technical and less relevant material to the appendices (e.g. discussion of comets, results for the CSP- and RAPA- treated cells) and reformulated some sections of the paper to clearly highlight when the text refers to hypotheses concerning the biological interpretation of our results. Our goal was to focus the manuscript primarily on the discussion of the Rd (α) and Sd (β) subpopulations.

Our response to your previous question on page 10 provides the line numbers in the new manuscript where these changes occur.

7. It is unclear why stress response pathways are being assessed in untreated cells without any stress exposure. Should this be analyzed in the drug-treated cells instead?

We have modified the text to only discuss pathways relevant to untreated (UT) cells in section 3 (which focuses on UT cells). In Figure 2 – source data 1 (list of gene signatures), we added a column entitled “UT relevant”, which indicates which of the 43 curated gene signatures is relevant for analyses associated with profiles of UT cells. Note however that there is evidence in the literature that some stress pathways do have (at least slight) differential expression across different cell cycle states. This includes, for example, heat shock proteins. This is reflected in the data where the signatures corresponding to these different stress pathways are also differentially expressed (for example, the oxidative stress response). [Changes L188-197]

8. In the growth curves in Figure 3 Supp 1a, it seems that both untreated and drug-treated cells take 2-3 days to reach stationary phase. This seems very unusual for cells that replicate quite rapidly under many growth conditions. Is this due to a nutrient-limited media or some other explanation?

We apologize for this confusion. Indeed, the cells were grown in YPD media so there should not have been any nutrient limitation per se. Although the underlying growth experiments had been repeated several times, we were concerned that there may have been some error in the measurement. Thus, we repeated these experiments and found different results. We now replaced these figures with the new data (Figure 3 figure supplemental 1B; Figure 5I); indeed the stationary phase is reached much earlier, and drug-treated cells exhibit slower growth as expected. We suspect that the confusion was due to a failure to account for the dynamic range of the spectrophotometer.

9. It seems surprising that extremely different antifungals (cidal vs. static, different cell targets) elicit a very similar transcriptional response based on this analysis (line 345-348). Can this be explained?

To be clear here, we observed that the majority of caspofungin (CSP) survivors have similar transcriptional profiles to FCZ cells in the α*/Rd* state; we did not see any CSP survivors with profiles resembling the β/*Sd* state, so there are some similarities and some differences between CSP, which is fungicidal in *C. albicans* and FCZ, which is fungistatic. Of note, because CSP is fungicidal, we used subinhibitory CSP concentrations (to ensure that a sufficient number of cells survived for profiling) and the CSP sc-profiling was only done at day 2. Nonetheless, we did examine cultures at later time points and with higher concentrations of CSP using microscopy. We always observed high levels of cell-cell aggregation which makes them inadequate for sc-profiling, yet did not observe any indication that CSP survivors of type β/*Sd* under any conditions (see eg Appendix 2 – figure 1E depicting Hsp70 fluorescence, marker of β/*Sd* cells, at day 2-3 for CSP-treated cells). In summary, perhaps cells at early time points post-treatment with low doses of CSP primarily cope with protein aggregation, a property shared with FCZ, but not with other tested compounds such as Rapamaycin.

We have added text to the Discussion section on the CSP response L428-435 and additional text regarding the RAPA response (which conversely only induces β/*Sd cells* L418-427).

10. Section 8-9 on expanding the analysis to day 6 was difficult to interpret in terms of the rationale for the experimental design and how the findings can be interpreted. It is also unclear how solid agar media-based tolerance assays can be extrapolated to liquid media growth assays with drugs, as these are very different conditions. It is also unclear if there is a day 6 untreated control that is being assessed.

Indeed, agar assays (disk diffusion assays, DDAs) and liquid assays (broth microdilution assays) do not give exactly the same results. This is because, in liquid cultures, cells grown in the presence of a drug can out-compete those cells that fail to grow in the drug. Nonetheless, when assayed at 48 hours, both disk assays and broth microdilution assays can report on tolerance (Rosenberg et al. 2018). Usually tolerance is measured at 48 hrs for cultures that grow at normal rates, and sometimes at 72 hrs for slow-going cultures. The reviewer is correct to question the meaning of the DDAs presented in Appendix 1 at day 6: whereas the cells harvested for sc-profiling at day 6 are return-to-growth without drug, the 6 day measurement in the DDA has not received replenished media. Nevertheless, the DDAS of Appendix 1 still indicates that the cultures do exhibit drug tolerance in the 2-3 day window relevant to our sc-profiling and that we do not observe drug resistance. We have added text in Appendix 1 to clarify this. [Changes L122-131 and in Appendix 1 L782-820].

a. The microscopy in associated Figure 4 K is very difficult to see and lacks scale bars.

We have decided to remove this panel; the two microscopy images were primarily there to show that we observed cells at the day 6 time point. The images do not advance the main findings in the manuscript significantly.

Reviewer #3 (Recommendations for the authors):Below I provide a few suggestions for how the authors may improve the manuscript, but overall I found it well done.1. First, it is my understanding that tolerance is defined as survival, but not growth, after exposure above the MIC for drugs that normally would kill most of the population of cells (for non-static drugs). This is the definition put forward in Balaban et al. ("Definitions and guidelines for research on antibiotic persistence"; Nat Rev Microbiol; 2019). I would have called what the authors describe as phenotypic resistance, i.e., a sub-population of cells that can continue to grow after exposure above the MIC, and a non-genetically heritable state (i.e., readily reversible). This is a known phenomenon for some antimicrobials/species combinations, as the authors describe. I think it would be good to standardize usage of the word tolerance through the manuscript, or the authors can better explain how the phenomenon they observe is consistent with Balaban et al.'s definition of tolerance. This is an important semantics issue.

Indeed we are discussing a phenotypic heterogeneity, (which is due to slow growth during constant exposure to a static drug); reviewers of prior work that characterized this phenomenon insisted that it be termed `tolerance’ because some older studies had referred to it in this way. Thus, although we suggested using a different term (perseverance) to distinguish this phenomenon from bacterial tolerance (which indeed is due to survival and not growth during periodic exposure to a cidal drug), editors of that paper (Rosenberg et al. 2018) agreed with the reviewers, and thus we are stuck with as suboptimal term which has different meanings in different contexts. For this manuscript, we tried to clarify this issue in the introduction (and to point out that the definition differs from that in bacteria treated periodically with cidal drugs). [Changes 62-73]

Second, the drug concentrations used were at or well-below their MICs. This raises the question of whether the authors are studying the clinical phenomenon that they describe as tolerance (i.e., above the MIC) when the experiment was conducted below the MIC. While I understand their reasoning for using low drug concentrations, and I think there is still something to be learned at these concentrations, I think the authors should better contextualize the clinical relevance of their findings with the caveat that the experimental results were probably collected at sub-clinical concentrations. Admittedly, drug concentration is dynamic during treatment, so at some point the concentration in vivo likely passes through the author's choice of in vitro concentrations. It would be interesting to know how much their results are invariant to the drug concentration chosen.

There appears to be some misunderstanding: the FCZ MIC_50_ of *C. albicans* SC5314 is 0.5-1 µg/ml. Here we used 1µg/ml fluconazole which is above the MIC (1-2x MIC_50_). We had previously stated ~1x MIC_50_ as this was an estimate based on MIC_50_ obtained from the literature. We have added a broth microdilution experiment to better estimate the FCZ MIC_50_ for the specific strain used in this study. The result now presented in Figure 3 —figure supplement 1A for FCZ at days 2 and 3, and Figure 5I for day 6.

Indeed, antifungal tolerance occurs at higher drug concentrations as well. However, antifungal tolerance is often very similar at drug concentrations ranging from just above the MIC to concentrations far above the MIC (e.g., Rosenberg et al. 2018, Nat. Comm.; Yang et al., mbio 2023; Todd et al., 2023). The reviewer is correct that the concentrations used here are sub-clinical and we agree that it would be very interesting to look at this question in a series of drug concentrations. This is a major challenge however in and of itself, and the cost of profiling/sequencing samples across a range of FCZ concentrations and time points with suitable controls is prohibitive, and would require substantial funding. In general, we agree it would be very interesting to better understand the dynamics of fungal drug tolerance in in vivo models relevant to the clinical setting and we hope to do so in the future.

2. The author's interpretation of the "comets" is questionable. On L244 they say, "This pattern suggests that the small set of cells from each comet have strong transcriptional similarity but each such comet is transcriptionally distinct from the other comets." I do not see how the cells within comets share "strong" transcriptional similarity because they are still spread out. Rather, the comet tails could be interpreted as lineages of descent, wherein each cell becomes more distinct (along some transcriptional axis) than its ancestor. This pattern is commonly observed in genetic data plotted with PCA, which is somewhat related to what is being shown here. Therefore, an alternative interpretation is that some lineages are moving toward a transcriptionally distinct state and leaving behind progeny along their trajectory, especially given that the experimental setup may not quickly remove ancestral cells in the two day exposure. Another explanation is that the comets are composed of a different (minority) morphology, such as filamentous growth. Another (less?) plausible interpretation is that these are noise clusters due to the inherent stochasticity involved in single-cell analyses. I see no reason why these explanations aren't equally reasonable to the author's explanation. Perhaps the authors can incorporate these alternative into their interpretations, but (at a minimum) there needs to be more justification given for the claim of "strong" intra-cluster transcriptional similarity.

We thank the reviewer for these insights. Because we do not consider the “comets” to be central to the main findings of the manuscript, and because we think that they potentially detract and distract from the central points, we decided to move the section on comets to the Appendix 2. In addition, we modified the text concerning comets to clarify the previous presentation [Changes L837-897]. Below we provide specific responses to the points raised above concerning interpretations of the comet data.

"This pattern suggests that the small set of cells from each comet have strong transcriptional similarity but each such comet is transcriptionally distinct from the other comets." I do not see how the cells within comets share "strong" transcriptional similarity because they are still spread out.

There are 19 smaller clusters or “comets” outside of the 5 major clusters analyzed in the main body of the manuscript. Each comet cluster is, in turn, comprised of a set of 7 to 126 cells. The cells within each comet are similar at the transcriptional level – the UMAP embedding procedure groups them together based only on their transcriptional profiles. However, the 19 different comets are not necessarily transcriptionally similar to one another, as noted, because “they are still spread out” and found in distinct clusters. [Changes 837-839]

Nevertheless, when we pooled all 19 comets and computed the list of all genes that were differentially expressed in the pooled comets relative to the 5 main clusters, we did find some interesting genes and pathways that suggest that these “outliers” are not simply technical artifacts of the DROP-seq profiling or the UMAP clustering algorithm. In particular, we discuss the ILV pathway which is over-represented in this group and its relationship to amino-acid starvation amongst other pathways revisited below.

We appreciate the original presentation of the analysis was dense and thus, we rewrote the section to better explain the use of the different tools and statistics. The comets are small and the sc-profiles are sparse, so analysis here really benefits from alternative methods. We have tried to better delineate these different sources of evidence [e.g. italic subtitles for each method on L844, 854, and 882]. Fortunately the different approaches largely highlight the same take home message, which we have tried to clarify in the presentation. [Changes L837-897]

We also removed the somewhat poetic language of “radiating from the centre of the UMAP”. We fully agree that this sort of “trajectory” is often reflective of some sort of model underlying the decomposition technique whether PCA (which is linear) or UMAP (which could be non-linear). This “linear radiation” first suggested to us that the comets might be some sort of technical artifact, but the differentially expressed genes and the pathway enrichment analysis suggests alternative explanations that have some support in the previous literature of *C. albicans* drug tolerance and its potential relationship with the genomic instability observed in *C. albicans*. [Changes 267-269]

We caution against interpreting the linear trajectories as true “evolutionary” trajectories. This is certainly one possibility, but the size of each comet and the current limitations of sc-transcriptional profiling would make a rigorous statistical assessment of this very difficult.

That the comets could be due to stochastic noise is another option, but we disagree that it is equally likely, given that both the differentially expressed gene analysis and the pathway analysis are tools underpinned by sound statistical models, and both provide evidence (at reasonable p-values or confidence intervals) that there is enrichment in specific functional processes. It is more difficult to estimate a formal confidence in the mapping of reads to chromosomal position (i.e., cluster 16 to chromosome 2) however it is at least visually convincing that ~38% of all reads from cells in cluster 16 mapped to this chromosome. We also note that comets are almost exclusively observed after FCZ treatment (and not after Rapamycin, for example). We have added a statement to the end of the section that all of our findings should be taken with caution due to the small sample sizes. Changes L889-890.

It is possible that some of these clusters correspond to filamentation, although (i) we filtered out large filamentous cells before profiling, (ii) filamentation signatures are not identified in the analysis, (iii) microscopy analysis of the cultures before filtering identified <1% of cells in a filamentous state. (Please see L148-149 and L492-493 in the manuscript.) It is possible that these cells could be representative of “small” germ tube cells, although we did not identify marker genes related to morphological switches.

This trifecta of analyses is why discussion regarding the comets remained in the manuscript – the enrichment of processes established to be central in fungal drug tolerance was far too high for this to be explained by “noise” (or the background models of any of these tools).

Our text was confusing (e.g. nature of intra- and inter-comet variability) and perhaps did not highlight why the extensive alternative analyses led us away from the idea that these were mere technical artifacts. However we agree that we are not able to fully characterize the comets at this point (e.g. exploring rigorously through DNA-seq what chromosomal aberrations exist and whether they line up well with the seminal work of Todd and Selmecki) so it remains descriptive at this point. We have minimized the discussion about DNA level aberrations (we also removed a paragraph about whole genome DNA profiling of the samples from the Discussion to simplify the manuscript).

3. L434: I am skeptical of the author's interpretation here about the relative fitness of α and β states. It could also suggest switching rates between phenotypic states are different between α and β populations. Also in this paragraph (L439), the authors use OD600 to show a higher growth rate for β over untreated. However, β also reaches a higher yield, which is unexplained. Having done many growth rate experiments with OD600 data, I am always skeptical of drawing large inferences about fitness from OD600. A better test of fitness is a competition experiment. I suggest the authors remove or downplay their conclusions about relative fitness here. Also, why isn't OD600 plotted in log-space (log(OD600)) if the goal is to see the difference in growth rate?

We fully agree with the skepticism regarding the relative fitness of α*/Rd* and β/*Sd* states, and these comments motivated us to revisit our group’s previous discussions as to what we can say regarding fitness and growth rates of the α*/Rd* and β populations. First, it is important to clarify that L439, which previously referenced Figure 4I (now Figure 5I) does not show β/*Sd* cells over untreated cells. The survivors at day 3 are a mixture of α*/Rd* and β/*Sd* cells. However, we did report that FCZ day 3 survivors grew faster after receiving new medium from days 3 to 6 compared to untreated cells, an observation that we struggled to explain. We decided to repeat the growth assays analysis of UT cells and FCZ day 3 survivors (Figure 5I) with care to ensure that both populations were diluted to the same OD_600_ initially (Methods 1C). (See response to Reviewer 2 Question 8). After correcting some prior issues with dilution and the dynamic range of the spectrometer, we now observe the more intuitive outcome where UT cells grow faster than FCZ survivors. Our previous manuscript stated that “At these concentrations, FCZ treatment slows growth relative to UT controls in the first days after exposure” which is now more visible with growth curves fitting what you would expect. We are sorry for this confusion.

Currently, we do not have a way to directly test the fitness of the α*/Rd* versus β/*Sd* survivors. We do comment however on the relative sizes of the α*/Rd* subpopulations compared to β/*Sd* subpopulations. In particular, from day 2 to day 6, we see that the β/*Sd* becomes more prevalent (L359-367). We do not have a mechanistic explanation but we offer at least two different models that are both consistent with our data as to why this may be the case (Changes L364-367 and L393-399, L413-417).

4. The authors did not seem to analyze feature importance in Leiden clustering, preferring to look at cluster associations (z-scores) instead. Leiden clustering is not guaranteed to be optimal, so some understanding of cluster stability would have been useful. That is, if inputs changed somewhat (e.g., subsampling or resampling), would the identified clusters have been similar? This is particularly important given that the UMAP1 versus UMAP2 clusters look like blobs that are split along arbitrary axes. It seems doubtful these clusters are stable, yet the whole manuscript analyzes them as though the blobs are discretized correctly.

This is an excellent point. We now include an updated analysis that incorporates measures of cluster stability throughout the manuscript. Before describing our changes and additional analyses, we want to note that the vast majority of single cell transcriptomic studies are performed with mammalian tissues. The diversity of cellular lineages and cell types (and extreme disruptions of normal physiological states in studies related to disease) induce UMAPs with distinct, non-overlapping clusters. We hypothesize here that this degree of separation will not be observed in a fungal population where the cells are isogenic and different only in their response to environmental cues. For example, we might expect that many cellular processes are similar across all individuals in the population at least in comparison to, for example, an epithelial versus a B-cell in the tumor microenvironment. Additionally, of course, fungal single cell profiling is in its infancy and better technologies that capture more transcripts per cell will likely produce more distinct clusters. Here, we hope that the reviewer agrees that we have provided evidence that the technology and analyses have succeeded to delineate, at least to some degree, between a handful of different types of cellular responses which we discuss individually below.

Additionally, we would like to clarify the nature of the methodology used here. We use a neural network (scVI) to process the raw data into a lower dimensional latent (hidden space). Leiden clustering is based solely on the expression profiles; the algorithm looks for evidence of separation between groups of cells. UMAP only maps the latent space (from scVI) down to 2D for visualization. Finally, pathway analysis used the identified Leiden clusters; this supervised analysis ultimately yields z-scores. As such, if a signature has higher expression in one Leiden cluster compared to other clusters, pathway analysis used in this manner is a form of cross validation: random clusters are not expected by chance to contain differentially expressed genes from the same functional categories. That is, the clusters point to interesting biological differences.

The microscopy depicted in Figure 2 C-E also represents a second source of cross-validation; it supports the separation of these clusters based only on GFP/RFP strains. It is therefore independent of the sc-transcriptomics data. Of course, this is not definitive and we agree fully with the reviewer that additional statistics can be used to examine stability which we present next.

We revisited the analysis of UT cells (this corresponds to Results subsection 3, paragraphs 1 and 2 L184-211). The reviewer’s question is primarily regarding the stability of the blue pink and green clusters in the clustering above on the right hand side (Author response image 2, this was Figure 2A in the original manuscript). Towards this end, we labeled every cell with a unique Cell ID (x-axis of the right hand side of Author response image 2). Then, we resampled 100x each time picking a subset of 95% of all UT cells each time, reapplying the scVI algorithm (to re-compute the model of expression) and reapplying the Louvain clustering algorithm (code available at https://github.com/vdumeaux/sc-candida_paper). We then determined which cluster each cell was found in. In the figure on the right hand side (Author response image 2), original cluster 1 corresponds to the darkblue/G1-S phase cluster in the left hand side image, cluster 3 corresponds to the green/M phase cluster and cluster 2 corresponds to sampled cluster labelled `1-3’. Some cells that were placed in cluster 1 originally again find themselves in cluster 1 in the resampling (that is the light purple pattern at the top right). Other cells are placed in cluster `1-3’. This means that, although there is good evidence that cluster 1 exists, the Leiden clustering is far from perfect and there is some confusion with clusters 2. This is also true for original cluster 3. Sometimes these cells are re-classified as cluster 3 but sometimes they are grouped again with cluster **`**1-3’.So although the three clusters are certainly not absolutely distinct there is strong evidence for all three clusters.

To better address the instability of some clusters within the manuscript, we re-represent our results so that they do not rely on the grouping of cells into discrete clusters. We produced an alternative to Figure 2 (page 8 of new manuscript) which shows the signature scores for each cell individually. For these signatures, we can observe that cells differentially express these signatures and that when a cell expresses a specific signature highly, it also tends to express other signatures highly. For example, a set of cells (cell indices >3000) exhibit elevated expression of genes involved in M phase, the heat-shock response and glycolysis.

Again we assigned all cells a unique Cell ID. They are ordered from left to right in the same manner across all 6 plots, providing the ability to compare them more readily, and to better highlight the differences of expression for specific signatures/biological processes. The text in the subsection describes essentially the three sets of cells (Index 0-2700, 2701-3000, and 3001-5000) and their inter-relationships. Note that here the analysis w.r.t. UT cells no longer depends upon the Leiden clusters.

Perhaps not surprisingly, the Leiden clusters identified post drug treatment (Figure 3, page 11 of new manuscript) are far more stable when we apply the same resampling technique. Figure 3 – Supplement figure 1C has been added to the manuscript. It shows that the 5 main clusters are quite stable (the 19 comet clusters were not included in the results due to their small sizes and given that we considered them both as one pooled comet and as distinct clusters in our analyses).

Figure 3 —figure supplement 1C was included to depict the considerable stability of the clustering that included untreated (UT) and drug-treated cells.

Figure 5B was also added to characterize the stability of clusters identified when investigating FCZ day3 survivors at day 3 and 6 after resuspension in fresh media. Most identified clusters are robust except for cluster 1 and 2 which includes α/Rd cells at day 6 and day 3, respectively. The lack of robust transcriptional differences between cluster 1 and 2 is described L347-350 and confirms that day 6 α/Rd cells likely evolved from day 3 α/Rd cells.

5. I have some concerns about the comparison between untreated and treated at the same (or similar) time points. If the transcript profile changes over time, as presumably it would, then it is reasonable to expect that the treated and untreated cells are at different equivalent times because their growth rates are different. I have a similar concern with comparing different time points (days 3 versus 6) because fresh medium (YPD) was used in this experiment. How much are transcript abundances static over time? This should minimally be discussed further in the manuscript. Also, the conclusions in L472-475 and L579-592 may need this caveat mentioned.

We agree that comparisons between UT and drug-treated populations are problematic. However we do not rely on direct comparisons between UT and drug-treated cells. Rather, analysis of the UT cells was performed to validate the efficacy of the sc-transcriptomics assay and provide a ‘sanity check’.

Unfortunately, the previous version of the manuscript had several sources of confusion with respect to the growth rate of the different *C. albicans* populations (for example Reviewer #2, Questions 4 and 8), and we have revisited the Discussion points with these criticisms in mind. Briefly, we better emphasize that the primary goal is not to compare UT with drug treated cells and that the major finding of the manuscript is the existence of the ⍺/*Rd* and β/*Sd* subpopulations initially identified in FCZ at day 2. This result is independent of any other time point or drug treatment (including untreated). The manuscript focuses on the observation that this same split between ⍺/*Rd* and β/*Sd* cells continues to exist at day 3 and even at day 6. While it is possible that some of the distinction between ⍺/*Rd* and β/*Sd* cells is not entirely due to drug treatment, it also cannot be due to cell cycle or nutrient levels alone and was not observed with other drug treatments (e.g. CSP and RAPA).